# A Qualitative Study into Teacher–Student Interaction Strategies Employed to Support Primary School Children's Working Memory

**Simona Sankalaite** [1,*], **Mariëtte Huizinga** [2], **Sophie Pollé** [1], **Canmei Xu** [1], **Nicky De Vries** [2], **Emma Hens** [1] and **Dieter Baeyens** [1]

[1] Parenting and Special Education Research Unit, Faculty of Psychology and Educational Sciences, KU Leuven, 3000 Leuven, Belgium; sophie.polle@kuleuven.be (S.P.); canmei.xu@kuleuven.be (C.X.); emma.hens@kuleuven.be (E.H.); dieter.baeyens@kuleuven.be (D.B.)

[2] Department of Family and Education Studies, Faculty of Behavioral and Movement Sciences, Vrije Universiteit Amsterdam, 1081 BT Amsterdam, The Netherlands; m.huizinga@vu.nl (M.H.); a.n2.de.vries@vu.nl (N.D.V.)

[*] Correspondence: simona.sankalait@kuleuven.be

**Abstract:** The current qualitative study examined the teacher–student interaction and its influence on children's working memory in primary schools in Belgium and the Netherlands. Eighteen primary school teachers participated in semi-structured interviews focusing on strategies employed to support students with working memory difficulties. The study offered a comprehensive overview of the strategies, categorised into instructional support, classroom organisation, and emotional support (based on the Teaching Through Interactions framework) that teachers use when dealing with working memory-related difficulties. Additionally, it provided unique insights into teachers' underlying beliefs and rationales about the effectiveness of these strategies. Lastly, factors influencing the use and efficacy of these strategies (based on the Multilevel Supply–Use model) were explored. By integrating teachers' voices and experiences, this research provides a unique opportunity to bridge theory and practice, and enrich the current understanding and interpretation of the teacher–student interaction and its implications for improving working memory performance in primary school students. Overall, the holistic approach, taking into account both direct and indirect approaches, offered a comprehensive understanding of the multifaceted challenges faced by students with working memory difficulties and the diverse strategies teachers can employ to address them, which can further inform classroom practices, professional development, and policy-making.

**Keywords:** working memory; teacher–student interaction; primary school; interview; qualitative study



## 1. Introduction

Working memory (WM) involves holding, manipulating, and transforming verbal and visual information. WM is one of the main executive functions (EFs) and entails a collection of cognitive processes that temporarily retain information in an accessible state, suitable for carrying out various mental tasks [1,2]. In addition, WM is considered a cornerstone of "higher-order" EFs—more complex cognitive operations, such as problem solving, reasoning, planning, and decision making (for a review, see [3]). Furthermore, WM is regarded as crucial for early academic success in reading comprehension and arithmetic (for meta-analyses, see [4,5]). Notably, in primary education, when foundational skills are developing, the function of WM becomes increasingly pertinent [6]. Although less extensively studied, contrasting with the extensive body of the literature on parents, research points to the importance of early teacher–student interactions (TSIs) on children's WM development ([7–9]; for a meta-analysis, see [10]). A recent review [11] examined

the causal effects of TSI strategies aimed at improving children's EFs in general and WM in particular. The authors concluded that the largest effects of employed manipulations were found in children considered vulnerable or disadvantaged. Such findings suggest that cognitive deficits can be minimised through appropriate and tailored support. Even though somewhat supported by theoretical frameworks and a number of studies, the efficacy and application of specific teaching strategies (rather than full-scale intervention packages), in practice, are explored considerably less [12]. Investigating and integrating teachers' perspectives can further bridge theory and practice by enriching the current understanding and interpretation of the TSI and its implications for improving WM. The current study, therefore, aims to provide a comprehensive overview of TSI strategies implemented by teachers in primary school classrooms, particularly focusing on children with WM difficulties and related problems.

### 1.1. Working Memory Development and Conceptualisation

EFs refer to various cognitive processes critical for goal-driven behaviour, especially important in novel, demanding situations [2,13]. Their significance to children's social and academic development is well-established [14,15]. The core EFs comprise inhibitory control (the ability to control attention, behaviour, thoughts, and emotions, which allows one to resist impulsive behaviours and ignore distractions), cognitive flexibility (the ability to adapt and switch between tasks or thoughts, which allows adjusting to new rules or priorities), and WM (the ability to temporarily hold, update, and manipulate information) [2]. Research has emphasised that WM may be one of the EFs particularly important to social functioning regarding peer rejection, social competence, and conflict resolution skills [16], as it plays a role in concentration, reasoning, decision making, and behaviour, among other things. Furthermore, out of all EF subcomponents, WM is the most predictive of academic outcomes and an even better predictor than intelligence quotient (IQ) scores (for a review, see [17]; for a meta-analysis, see [4]).

WM starts developing in infancy and rapidly develops in preschool [18,19], with pronounced changes occurring up to the age of 9, with modest changes in age effects across ages 9 and 11 [20,21], however, slowing during adolescence [22]. The development of WM parallels developmental changes in brain structures, like the maturation of the prefrontal cortex, synaptic pruning, and myelination [23]. Based on the findings from behavioural and neuroimaging research, (early and middle) childhood seems to be a period characterised by plasticity, sensitivity, and responsivity to developmental and environmental influences [24–28].

Two seminal models were proposed to describe and conceptualise WM [29,30]. Both models agree that WM comprises several components or processes that operate in a coordinated manner to temporarily store, manage, and manipulate information; however, they have somewhat different emphases. The first, Baddeley and Hitch's model [29], consists of a central executive (a control system of limited attentional capacity) and two temporary storage systems: the phonological loop (related to sound and language; for the passive storage and active rehearsal and maintenance of information) and the visuospatial sketchpad (oriented towards visual stimuli; for the storage and rehearsal). These components were later augmented by the episodic buffer, which provides temporary storage in which various components of WM interact and interface with information from perception and long-term memory [31,32]. The second, Cowan's [30,33] proposed model, offered a somewhat different perspective. Cowan's embedded-process model comprises four elements: central executive (circulates information that is within the focus of attention), long-term memory (repository of information), activated memory (information that is currently activated and accessible), and the focus of attention (a limited amount of information that is subject to immediate control and manipulation). Cowan defines WM as a set of cognitive processes that act on long-term memory guided by attention, such as retrieval and maintenance [34]. The idea of the embedded-process model is to provide a more general description of the WM system by describing the processes it involves rather than to divide its components

based on the form of stored information, as in the multicomponent model. Both models highlight the complexity of the WM construct, which is important to consider when identifying unique WM-related difficulties, addressing WM's role in learning, and tailoring the support provided.

*1.2. Working Memory and Education*

WM plays a crucial role in classroom interactions, learning, and the execution of complex school tasks. WM is involved in processing and understanding new information, following instructions, and remembering and applying knowledge [35–37]. Furthermore, WM is involved in language processing and communication as it helps children remember and comprehend sentences, follow conversations, and generate coherent responses [38], which is important for social and academic development.

There is ample research addressing and describing the contribution of WM to academic achievement. Cross-sectional studies investigating the relationship between WM and academic performance have consistently shown that students with better WM capacities tend to have higher academic scores (for meta-analyses, see [4,39]). Observational studies exploring the associations in naturalistic educational contexts have documented the important role that WM plays in classroom learning and academic achievement, (e.g., [40]). A number of longitudinal studies provided compelling evidence for the predictive role of WM in academic outcomes, showing that children's WM capacities in preschool or early childhood can predict their later academic, especially maths and literacy, achievement [41–43]. Furthermore, WM can also be viewed as a mediating factor that influences the relationship between other cognitive skills and academic performance, (e.g., [44]). For example, WM mediates the relationship between attention control and reading comprehension; that is, attention control impacts WM, which, in turn, affects reading comprehension. Finally, WM can also act as a moderator, (e.g., [45,46]). For instance, the impact of a specific teaching strategy on academic performance might depend on a child's WM capacity. Here, WM acts as a moderating variable, influencing the strength or direction of the relationship between instructional strategy and academic performance. Thus, WM impacts academic achievement in different ways, both direct and indirect.

However, the conventional concept of a unidirectional association between cognitive abilities and academic performance has recently been challenged. Instead, the mutualism theory proposes that various skills and competencies foster a bidirectional relationship during the course of development. This reciprocity emerges from mutually beneficial interactions among cognitive processes initially perceived as unrelated [47]. Notably, novel studies [4,48–51] have demonstrated the presence of a bidirectional and longitudinal influence among these variables [52,53].

Given the importance of WM to children's academic outcomes and social and emotional development, there have been numerous attempts to improve and strengthen WM in primary school children [54]. Intervention studies suggest a number of diverse practices and programmes for effectively increasing children's WM. Most interventions aim to improve WM directly and involve tasks designed to challenge and, therefore, strengthen WM. Such interventions include a variety of different exercises increasing in difficulty that target both verbal and visuo-spatial WM and, through repeated, targeted practice, increase WM capacities, (e.g., [55,56]). However, the majority of studies, directly and exclusively training WM, often fail to report the durable effects. Some training programmes lack transferable results [57–59] and fail to generalise to academic outcomes [60,61]. Other interventions produce short-term training-specific effects that, unfortunately, do not generalise and decline shortly after the intervention is completed (see meta-analyses by [62–64]). This lack of transferable and durable effects might be attributed to such interventions training WM out of context and, therefore, ignoring potentially important contextual factors. Finally, recent reviews (see [65,66]) concluded that evidence that WM training leads to improvements in other areas, such as literacy and numeracy, is inconclusive at best, and further research is

required to better understand the mechanisms through which WM training may improve WM and academic performance in children.

A growing body of evidence suggests that indirect approaches may also be highly effective, especially when complementing direct training [67–69]. Such approaches aim to enhance WM not by directly training it, but rather by addressing other skills or factors that can influence it. For instance, these interventions might focus on teaching WM-supporting strategies that help children manage the demands placed on their WM. These strategies include chunking or breaking down information into smaller, more manageable parts, using rehearsal strategies (e.g., silently repeating information to keep it active in WM), or visualisation strategies to create mental images of information. These approaches might not directly improve WM but, instead, aim to adapt the learning environment and teaching methods to reduce WM demands placed on the children. Such adjustments can help children with weaker WM perform and succeed in the classroom without needing to improve WM capacities [70–72].

Despite the well-established significance of WM in children's social and academic development, there are still gaps in the current understanding of how to effectively and durably improve WM. Given the dynamic interplay between cognitive processes and the environment, the TSI stands out as a pivotal contextual factor, shaping the immediate learning environment. It is increasingly recognised that indirect strategies, addressing factors influencing WM or adapting learning environments to accommodate children with differing WM capacities, may offer a promising avenue for enhancing children's WM and, in turn, academic performance. Further research is required to understand the specific contextual elements and the best combination of teaching strategies that effectively optimise WM and, consequently, positively impact children's learning and overall academic development.

### 1.3. Teacher–Student Interaction

Considering that primary school-age children spend a significant amount of time in school, the nature and quality of their interactions with teachers have gained increasing attention, especially when exploring their impact on children's educational outcomes. Children interact with teachers on the classroom level (i.e., the TSI) and the dyadic level (i.e., the teacher–student relationship). Regarding interactions between teacher and student at the classroom level, a significant contribution was made by Hamre and Pianta [73]. The researchers introduced a Teaching Through Interactions (TTI) framework (see Figure 1), organising the TSI into three broad domains representing distinct areas of these interactions. Namely, instructional support—for instance, asking open-ended questions; classroom organisation—for instance, clarifying the rules and expectations; and emotional support—for instance, acknowledging children's emotions and experiences, and sensitively responding to them [74,75]. Furthermore, the framework divides the three domains into specific dimensions—subcategories, each reflecting a unique aspect or quality of the TSI domains. Finally, these dimensions are further subdivided into indicators—the concrete manifestations or observable behaviours, actions, or elements of the TSI that fall within a specific dimension. The proposed framework places the TSI as a cornerstone for student learning. This framework suggests that the most important for learning might not be what teachers teach but, instead, how the teachers engage with the students. More specifically, the ongoing, moment-to-moment interactions between teachers and students form the foundation for student achievement and social, academic, and cognitive development. Research suggests that WM skills grow more during the school-year months compared to the summer months, suggesting that school environments provide children with unique opportunities to improve and practice their WM skills [76].

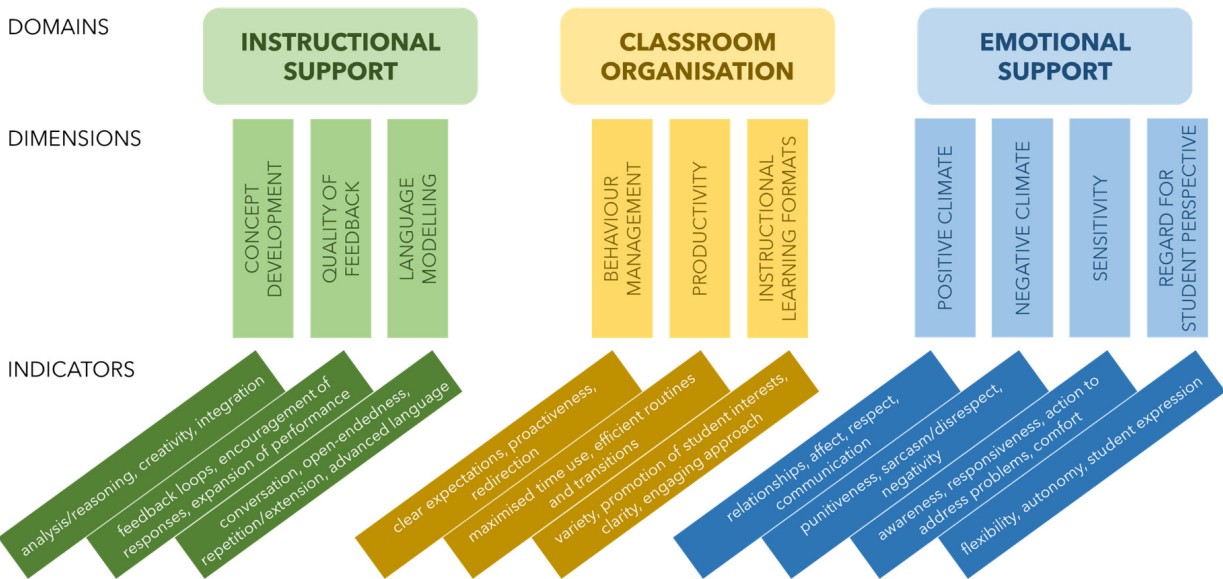

**Figure 1.** The Teaching Through Interactions Framework adapted from [75].

Each of these domains contributes to the overall learning environment and supports cognitive processes, including WM. Various theories have been proposed as potential explanations for why and how the TSI supports WM. Instructional support most strongly aligns with Vygotsky's sociocultural theory [77]. According to the sociocultural theory, learning is a socially mediated process. At the core of this theory is the idea of the "zone of proximal development", which describes the gap between learners' independent (i.e., without help) capabilities and their achievements with received assistance. Teachers, by scaffolding instruction, asking open-ended questions and facilitating meaningful discussions, aid students in internalising and constructing knowledge. Collectively, these strategies not only promote deeper understanding, but also support the optimal functioning of students' WM in the learning environment. The classroom organisation domain can be linked to Bandura's social learning theory [78]. This theory suggests that students learn from observing and imitating others' behaviour, attitudes, and the consequences of such behaviour. In a well-organised classroom, where expectations are clearly stated and rules are consistently enforced, students learn and internalise appropriate behaviours through observation and modelling. Additionally, classroom organisation helps in managing students' attention and ensuring smoother transitions between activities. These structural changes not only promote positive behaviour, but also support cognitive processes like WM. Over time, the role of the teacher as an external regulator decreases, allowing students to gradually move into more autonomous regulation in their learning. The emotional support domain is primarily supported by the attachment theory [79]. According to Bowlby's attachment theory, children develop their primary understanding of self and others based on their relationship with the primary caregiver. In the context of education, teachers, who provide emotional support, facilitate the creation of a supportive and nurturing learning environment where students feel emotionally secure, valued, and connected to their teacher. Such an environment, in turn, enhances students' engagement, motivation, and overall academic and socio-emotional well-being. Teachers' sensitive responses to students' emotions and experiences, and the emotional climate they establish in the classroom, play a crucial role in shaping students' learning experiences. When students feel emotionally supported, they are less likely to experience stress that could negatively impact their cognitive processes. Moreover, emotional support aligns with the principles of self-determination theory [80], which emphasises individuals' needs for relatedness, competence, and autonomy. Emotional support provided by teachers directly caters to these needs. Engaging students in active learning, building competence via skill-building and feedback, providing opportunities for choice, and acknowledging students' perspectives can foster these needs and, hence,

boost their intrinsic motivation, which, in turn, enhances their engagement and persistence in academic tasks. When students' emotional needs are met, they are better positioned to tackle challenges, absorb new information, and use their WM more efficiently.

Developed alongside the TTI framework, the Classroom Assessment Scoring System (CLASS; see Appendix A) [81] is a validated set of measures for assessing the quality of TSIs across various educational settings. CLASS closely follows the TTI framework by further separating three broad domains into multiple dimensions and corresponding indicators, which are then observed and scored. As a result, the CLASS transforms theoretical concepts articulated in the TTI framework and provides an operational tool that is instrumental in observing and quantifying the TSI, facilitating its application in professional development, teacher training, and research. In agreement with the TTI framework (and the CLASS), numerous observational studies have consistently demonstrated a correlation between the quality of TSIs and children's WM skills [82–85]. A recent meta-analysis [10] informed further regarding the strength of these associations, revealing small-to-medium overall effect sizes. Additionally, there are indications of causal effects of TSIs on children's WM (for a review, see [11]), with some studies suggesting that individual and contextual characteristics may influence the effectiveness of TSI strategies and their impact on children's outcomes [86,87]. This underscores the importance of considering the child and teacher's characteristics, and classroom context when providing effective support to students. Further exploration is warranted to identify influential characteristics and to better understand the interplay between such factors and TSIs, as well as students' WM.

A lack of studies exploring specific factors that influence effective teaching practices and interactions between teachers and students underscores the need for a comprehensive analysis of such factors. Brühwiler and Blatchford's [88] multilevel supply–use model (see Figure 2), highlighting the fundamental role of effective teaching practices and interactions between teachers and students in promoting positive learning outcomes, serves as a valuable framework. This model operates at four interrelated levels, each contributing uniquely to the learning outcomes. At the "supply" levels, the model focuses on the characteristics of the broader educational system, the context of the school (e.g., team), classroom context and processes (e.g., class size or quality of teaching), and teacher characteristics and competency (e.g., teaching experience or differentiated instruction). These factors collectively constitute the "supply" of educational inputs provided by the teacher to facilitate student learning. At the "use" level, the model examines how students engage with the multifaceted support provided by the teachers. Here, it delves into the intricacies of learning environments (i.e., peers), individual learning preconditions (e.g., cognitive or motivational cognitions), and processes (e.g., learning strategies, attention, and effort). The "use" level elucidates how students interact with the "supply" and the extent to which they harness these resources to enhance their learning experiences and outcomes (i.e., student achievement and competencies). While the model underscores the significance of various factors at different levels, further research is essential to understand their complex interplay and identify the most influential factors in fostering high-quality TSIs, ultimately leading to the most positive learning experiences and outcomes.

The scarcity of studies underlines the need for a comprehensive understanding of the specific strategies and mechanisms employed by teachers in the classroom. Qualitative research can prove to be particularly valuable for addressing this gap, as it excels in generating novel insights and uncovering information that quantitative methods alone cannot capture [87,89]. It is crucial to recognise the unique value of semi-structured interviews, which can unveil important dimensions and indicators not covered by the TTI framework, and lead to unexpected or unanticipated findings [90]. Through the interviews, teachers have the opportunity to share the strategies they employ, elucidate the challenges they face, and describe their adaptation to different students or situations. This insight into teacher experiences is essential to better understand the strategies teachers already apply in their classrooms and uncover where and how teachers can enhance their teaching practices through developing, refining, and implementing interventions aimed at improving the TSI.

Therefore, a combination of a deductive "top down" approach grounded in the TTI theoretical framework and an inductive "bottom up" approach rooted in teachers' practice-based experiences allows for insights into what the teachers are implementing in their classrooms. Furthermore, this combined approach can reveal additional aspects not covered by existing theories, which may hold particular or unique relevance when supporting children's WM. Additionally, drawing inspiration from the multilevel supply–use model [88] can shed light on the contextual conditions required for the effective implementation of these strategies. The model also highlights the significance of individual attributes belonging to both teachers and students, which are essential for deriving benefits from the applied strategies.

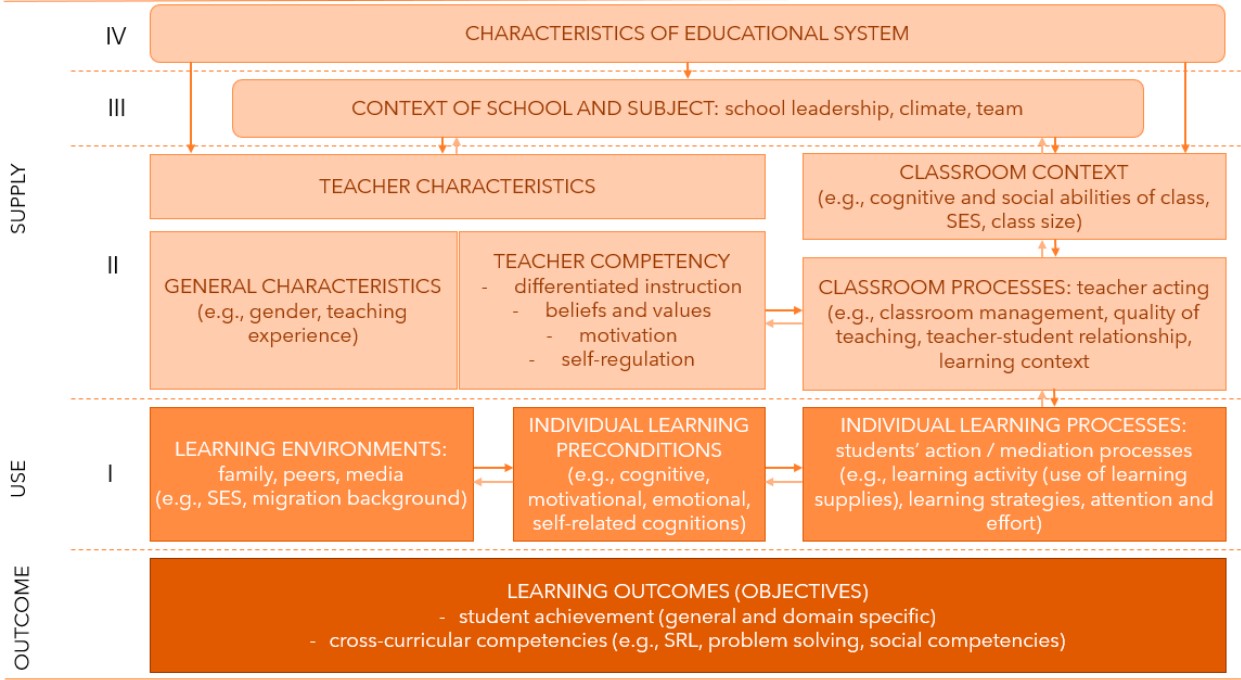

**Figure 2.** A Multilevel Supply–Use Model of Student Learning adapted from [88]. Note: I–IV refer to four interrelated levels: I—"use" and II–IV—"supply". The arrows indicate the direction of interactions between and within the levels.

### 1.4. Current Study

The current study will apply a qualitative approach—semi-structured interviews with experienced primary school teachers in Belgium and the Netherlands. This study is centred on addressing three key research questions. The first goal is to identify specific TSI strategies employed by the teachers to support their students' WM and manage WM-related problematic behaviour in the classroom. The second goal is to explore the underlying beliefs and rationales that teachers hold regarding the effectiveness of the identified strategies. Finally, this study aims to examine the factors and characteristics that influence the utilisation and efficacy of these strategies.

By integrating teachers' voices and experiences, this research provides a unique opportunity to bridge theory and practice, enrich the current understanding and interpretation of the TSI and its effectiveness, and aims to offer valuable insights that can inform classroom practices, professional development, and even policy-making.

## 2. Materials and Methods

### 2.1. Participants

This study was conducted in the Dutch-speaking region of Belgium and in the Netherlands. Data were drawn from interviews with 18 primary school teachers (10 Belgian and 8 Dutch). This amount of data is commonly considered sufficient to reach saturation [91–93].

Teachers were eligible for study entry if they taught in regular primary education and had at least three years of teaching experience. As the demographics of participants show (see Table 1), two (11.11%) male and 16 (88.89%) female teachers took part in the study. Participating teachers varied in years of teaching experience (M = 14.6, SD = 9.06, min = 3, max = 34) and held different positions, including teaching in one classroom (*n* = 13; 72.22%), teaching in multiple classrooms (*n* = 2; 11.11%), co-teaching in multiple classrooms (*n* = 2; 11.11%), and providing additional support to students with special needs (*n* = 2; 11.11%). In Belgium and the Netherlands, primary education (excluding preschool) consists of 6 years of teaching—years 1 to 6.

**Table 1.** Demographic Information of the Participating Teachers.

| Participant | Gender | Nationality | Years of Experience | Position |
|---|---|---|---|---|
| Participant 1 | Male | Belgian | 3 | Providing additional help to students with special needs and co-teaching in years 2 and 5 |
| Participant 2 | Female | Belgian | 27 | Teaching in year 5 |
| Participant 3 | Female | Belgian | 8 | Teaching in year 2 |
| Participant 4 | Female | Belgian | 10 | Teaching in year 5 |
| Participant 5 | Female | Belgian | 19 | Teaching in year 3 |
| Participant 6 | Male | Belgian | 6 | Teaching in year 6 |
| Participant 7 | Female | Belgian | 25 | Teaching in year 2 |
| Participant 8 | Female | Belgian | 20 | Teaching in year 3 |
| Participant 9 | Female | Belgian | 5 | Teaching in year 6 |
| Participant 10 | Female | Belgian | 34 | Teaching in year 6 |
| Participant 11 | Female | Dutch | 13 | Teaching in year 4 |
| Participant 12 | Female | Dutch | 14 | Teaching in years 5 and 6 |
| Participant 13 | Female | Dutch | 20 | Providing additional help to students with special needs |
| Participant 14 | Female | Dutch | 3.5 | Teaching in year 6 |
| Participant 15 | Female | Dutch | 20 | Teaching in years 1–3 |
| Participant 16 | Female | Dutch | 20 | Teaching in year 4 |
| Participant 17 | Female | Dutch | 12 | Co-teaching in years 1–6 |
| Participant 18 | Female | Dutch | 3.5 | Teaching in year 6 |

*2.2. Procedure*

The study protocol was approved by the Social and Societal Ethics Committee of KU Leuven (G-2019-1320). For recruitment, an invitation email explaining the aim of the project was sent to principals and secretaries of primary schools in Belgium and the Netherlands. Interested teachers received further information by email or phone call. The final sample consisted of 18 teachers from 15 schools. Interviews were conducted by the members of the research group. Depending on the preference of the teacher, interviews took place at school (*n* = 17; 94.44%) or at their home (*n* = 1; 5.56%). With the consent of the participants, all interviews were audio-recorded by the interviewer.

*2.3. Data Collection*

At the beginning of the interview session, each participant was informed about the goals and design of the study and was provided with an informed consent form to sign. The interview process was guided by questions based on the WM subscale of the Behaviour Rating Inventory of Executive Function (BRIEF) [94] and the TTI [75]. Semi-structured interviews started with a brief introduction to WM. Additionally, teachers were informed about three dimensions of TSI support (i.e., instructional support, classroom organisation, and emotional support) and instructed to consider all the dimensions when answering questions. The interviewer then presented nine examples to the teacher, which described a student exhibiting a WM-related problem behaviour in the classroom (based on the nine items of the WM subscale of the BRIEF). For instance, for the item "Your student

needs repetition of the explanation, otherwise it will not stick/get through", a situation "You have just given instructions on how to add fractions and the students should now work independently on three exercises. After a few minutes, the student comes to you with the question which exercises s/he had to do and/or how s/he should do it". The teacher was then asked to choose four situations that s/he was most familiar with and describe in detail the actions the teacher takes when dealing with WM-related problematic behaviour. For each of the four selected items, teachers were asked to provide an example of a situation that occurred in their classroom. Afterwards, TSI strategies used by the teacher to improve WM performance were discussed. During the conversation, open questions were asked regarding:

(1)　the TSI strategies used by the teacher to support a student with WM-related problematic behaviour (e.g., "If you think of emotional support, what do you do to help a student with WM problems?");

(2)　the teacher's reasons for using these TSI strategies for strengthening WM or reducing/improving WM-related problematic behaviour (e.g., "You mentioned that you give feedback, why do think that giving feedback helps to improve WM performance?");

(3)　the factors that may influence the effectiveness of these TSI strategies on reducing/improving WM problems (e.g., "Out of these strategies, are there any strategies that work better for some students than others, and, if yes, what are the characteristics of these students?").

The researcher then summarised the main insights and checked with the participating teacher whether these conclusions were correct. Finally, the interviewer thanked the interviewee for his/her participation. The interviews took approximately one hour (M = 57 min, SD = 7.17, min = 45, max = 67). The interviews were conducted in Dutch, consistent with the primary language of the participating teachers.

### 2.4. Template Analysis

To analyse the data, a thematic analysis was used, a method defined as "identifying, analysing and reporting patterns (themes) within data" [95] (p. 79). Data were analysed using a structured codebook [96]. A template analysis enables a combination of an inductive or 'bottom-up' way and a deductive or 'top-down' way of coding (see Figure 3). A theoretical framework to analyse the data was used to (allow the opportunity to) contribute to the revision of different aspects of the theory.

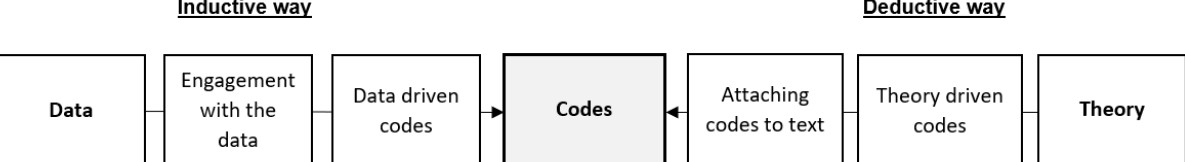

**Figure 3.** Inductive–Deductive Hybrid Approach for the Code Development adapted from [97].

In the template analysis, the researcher identifies themes important to the research question, codes these themes, and then organises them in a coding 'template' [98]. Below, six steps of the template analysis, as carried out in the current study, are presented.

Step 1: Transcribing interviews and getting familiarised with the data

The initial step involved transcribing the interviews and getting familiarised with the data. All interviews were transcribed verbatim by the members of the research team.

Step 2: Defining a priori themes

Secondly, a codebook was created by defining a priori themes based on the TTI framework [75] and the CLASS [81]. CLASS organises the TSI into three domains: instructional support, classroom organisation, and emotional support, which are further divided into a

total of 10 dimensions (e.g., positive climate within emotional support) and subdivided into 42 indicators (e.g., relationships, affect, respect, and communication within a positive climate). These 42 indicators served as a priori themes (labelled based on the CLASS conceptual framework, see Appendix A).

To analyse potential mechanisms through which TSIs may promote WM, the theories referred to in previous research were examined [10,75]. Consequently, a priori themes were identified from the attachment theory [79], social learning theory [78], sociocultural theory [77], and self-determination theory [80]. Themes identified from the central concepts of these theories are: (1) a secure base and safe haven for the attachment theory, (2) modelling and an external regulator for the social learning theory, (3) scaffolding, language development, and challenging learning activities for the sociocultural theory, and (4) competency, connection, and autonomy for the self-determination theory.

A multilevel supply–use model adapted from [88] (see Figure 2) was employed to analyse factors influencing the effectiveness of TSI strategies in supporting WM. This model encompasses four levels for describing the multiple conditions of student learning: the system, school, and teacher/classroom at the "supply" levels, and the individual student at the "use" level. Multiple factors are differentiated within the levels of the teacher and the student (e.g., learning environments and individual learning preconditions within the individual student level). These factors are further subdivided into 27 different elements, with a priori themes identified, categorised, and labelled across all levels.

To ensure the quality of a priori themes, initial discussions between the researchers were held regarding the approach to be followed. A codebook was developed based on the themes derived from the TTI framework, CLASS, proposed theories, and the multilevel supply–use model, with minor adjustments made after further discussion.

Step 3: Coding of the data template

In the third phase, all interviews were systematically coded in NVivo 12 [99]. Important data for answering the research questions were identified and assigned a priori themes. However, if part of the interview was not encompassed by one of the a priori themes, an existing theme was modified (e.g., a 'support for autonomy and leadership' theme was updated to a 'support for autonomy, leadership and taking responsibility' theme); if an existing theme could not be modified, a new theme was generated.

Following established guidelines for qualitative research [100], 22.22% (4 out of 18 interviews) were double-coded to ensure inter-rater reliability. Inter-rater reliability was assessed using Cohen's kappa (ranging from −1 to +1), with a resulting value of 0.97, indicating almost perfect agreement [101].

Step 4: Creating an initial template

Next, an initial template was produced by grouping codes into higher-order themes after coding all the interviews to avoid presuppositions. Categories or (sub)themes lacking data were removed (e.g., language development) to maintain accuracy and relevance.

To organise the TSI strategies, the TTI framework [75] and the three TSI support domains were used as higher-order themes labelled 'instructional support', 'classroom organisation', and 'emotional support'. A check was run to identify additional domains through data-driven coding; however, the TTI framework appeared to cover the interactions mentioned by the teacher. To group potential mechanisms, three broad higher-order themes were labelled 'TSI helps because it trains the WM', 'TSI helps because it supports the WM or relieves the demands of the WM task', and 'TSI works because it focuses on factors that influence WM performance'. Finally, to group the influencing factors, three higher-order themes named 'classroom and teacher level', 'individual student level', and 'context level' were used.

Step 5: Developing the template

In the fifth step, the initial template was refined and improved for enhanced effectiveness. This stage involved collaboration and discussions among the researchers. During

these discussions, some existing (sub)themes were modified, and new (sub)themes were added for the TSI strategies. The data extracts were then re-evaluated to ensure they aligned with the identified (sub)themes, and a few codes were assigned to a different (sub)theme. This iterative process aimed for the template to accurately represent the data. While developing the template, the identified mechanisms were grouped based on relevant theories (e.g., attachment theory and Vygotsky's sociocultural theory). Additionally, new subthemes emerged that were not tied to specific theories, becoming more general themes, like 'attention and focus' and 'repetition and routine'. Finally, the influencing factors were further categorised (e.g., individual student level into child characteristics, individual learning preconditions, and individual learning processes).

Step 6: Using the final template to help interpret and write up the findings

The updated coding template served as a valuable tool for data interpretation and guided the writing process (see Appendix B). This phase involved a thorough examination of the (sub)themes, leading to a deeper understanding of the data and the identification of recurring themes and patterns.

Firstly, the (sub)themes were revisited, focusing on key findings and notable insights from the analysis. Secondly, the relationships and connections between these (sub)themes and the broader framework were analysed to explore how they intersected and contributed to a better understanding of the research questions. For instance, the connections between autonomy and intrinsic motivation or the interdependency and interaction of the TSI domains were explored. Finally, quotes from the interview transcripts were selected to support and illustrate the findings. These quotes served as concrete examples that represented the participants' perspectives and experiences. The final template helped to identify relevant quotes that aligned with the specific (sub)themes, adding authenticity and richness to the research findings. Ultimately, the findings derived from the template analysis, along with the identified themes and patterns, were documented in a comprehensive write-up.

## 3. Results

The overarching objective of this study was to obtain a comprehensive overview of the specific strategies implemented by teachers in primary school classrooms, particularly focusing on children with WM problems and related difficulties. The findings are presented in a structured way, aligning with the three research questions (see Table 2 for an overview).

The first goal was to identify specific TSI strategies employed by the teachers to support their students' WM and manage WM-related problematic behaviour in the classroom. As outlined in the Introduction, four theories have been proposed by the literature (i.e., top-down approach) as potential explanations for why and how TSI supports WM. These four theories, (1) sociocultural, (2) self-determination, (3) social learning, and (4) attachment theory, are, therefore, used to group and structure the findings on teacher-employed strategies.

The second goal was to explore the underlying beliefs and rationales that teachers hold regarding the effectiveness of the identified strategies (i.e., bottom-up approach). If aligning with the literature, these rationales are presented under a corresponding theory by providing definitions pooled from various teacher answers. However, if teacher-identified rationale does not directly align with either of the four literature-proposed theories, and, instead, introduces overlap—the additional rationales are discussed on their own at the end of the section.

The final goal was to examine the factors and characteristics that potentially influence the utilisation and efficacy of these strategies. Such factors, as identified by the teachers, are described in line with the multilevel supply–use model, starting from the individual student level and broadening up to the system level.

**Table 2.** The Overview of the Specific Strategies Employed by the Teachers and the Underlying Theory/Rationale Based on the Literature or Identified by the Teachers and Linked by the Researchers.

| Literature | Teacher Interviews | |
|---|---|---|
| *Top-down approach* | | *Bottom-up approach* |
| Identified theory **based on the literature** | Strategy employed by the teacher | Rationale **identified by the teacher** |
| Sociocultural theory | Providing information | Scaffolding |
| | Increasing challenge | |
| | Adjusting support (individualised approach) | |
| | Monitoring | |
| | Critical thinking | Metacognition |
| | Prompts and cues | / |
| | Classroom arrangement | / |
| Self-determination theory | Varying modalities/materials | Attention and focus & (intrinsic) motivation |
| | Promotion of student interest | |
| | Providing feedback | / |
| | Autonomy, leadership, and responsibility (gradual release of responsibility) | / |
| | Student expression | / |
| Social learning theory | Providing an overview | Repetition and routine |
| | Routines | |
| | Rules | / |
| | Behaviour management | / |
| | Modelling | / |
| | Self- and parallel talk | / |
| Attachment theory | Emotional safety | Safe haven & secure base |
| | Attuned communication | |
| | Responsiveness | |
| | Sensitivity | |

Note: / signifies that rationale was not mentioned by teachers.

*3.1. Theory/Rationale and Strategies*

3.1.1. Sociocultural Theory-Based Strategies

The literature, in regards to the sociocultural theory, suggests that learning is mediated through social interactions—teachers scaffold instruction, pose open-ended questions, and foster discussions, guiding students to internalise and construct knowledge.

**Teacher's Rationale.**

*Scaffolding.* Scaffolding refers to temporary support by the teacher to help students with tasks that are initially beyond their capabilities. Through scaffolding, teachers gradually reduce the level of support as students gain independence in their learning. By breaking down complex tasks into smaller, more manageable chunks and providing clear, step-by-step instructions, teachers indicated that students could overcome the challenges posed by WM problems.

P14: *"Because then they have less to oversee what they need to do, so it becomes more manageable."*

In other words, teachers state that scaffolding allows a gradual building of knowledge and understanding, reducing the cognitive load, thereby enhancing students' engagement.

P17: *"If a child engages with a task in various ways, not just by listening but also by executing it and discussing it together, it's logical that children will stay engaged."*

Moreover, by offering challenging learning activities (e.g., articulating concepts and ideas in their own words/independently), students are given the opportunity to engage, train, and strengthen their WM, as indicated by multiple teachers.

P2: *"I challenge them to put it in their own words. Explaining something yourself is the highest form of learning. So, when you let children explain something themselves, they're still thinking about it because they have to verbalise it, they have to formulate it, and sometimes you see them regaining their thoughts, so they're thinking about it at that moment."*

**Strategies.**

*Providing information.* Fourteen teachers provide instructions by breaking them down into smaller, manageable pieces, guiding a student step-by-step to complete an exercise (as specified in the teacher-provided example). In the case of students with WM problems, it is crucial that the instructions provided are concise, straightforward, and simple. By keeping the instructions sequential and focused on one element at a time, children can better comprehend, concentrate on, and process the relevant information without getting distracted by irrelevant details.

P3: *"You shouldn't say too much to her at once, for example, 'You have to add 50 plus 70'. No, first, you have to start with 50; that's what we'll focus on. Then, the next step is addition. And what does that mean? It means adding. So, step by step, you need to work through it."*

Furthermore, seven teachers avoid overwhelming students by providing too many instructions or assignments at once.

P14: *"I say, 'You're going to do this now', period. Just give one task and repeat the explanation. Break it down into steps or ask the child, 'What is the most important thing you have to do?'. So, make a kind of priority list. Now you're going to do that for 10 minutes first, and then we'll see what we'll do next."*

Finally, when supporting children with WM problems, ten teachers structure the information provided. Excessive or disorganised information can lead to distractions and hinder students' focus on the relevant content.

P3: *"That's chaos, while it's very clear for us. Now we have to do addition exercises. For them, they already see a drawing there, they already see a different exercise, and that's chaos. They really need a teacher who provides that structure."*

*Increasing challenge.* Another effective strategy teachers use to enhance students' WM is the progressive building up of the difficulty level of tasks/assignments and slowly decreasing (i.e., fading out) the support or instructions provided, as outlined by six teachers. By doing so, they aim to challenge their students and, in turn, activate and train their WM.

P2: *"That's what I'm really practising now. To consciously give multiple instructions at once. In second grade, you'll say, 'Now I want you to first put that math sheet in your folder behind the red tab', and then you wait until everyone has done that. And only then do you say, 'Now go get your lunchbox'. If they're a bit older, you say, 'Put that sheet away and get your lunchbox'. The older they get, the more tasks there are."*

A teacher can gradually decrease the amount of support provided, which, by challenging children to search for answers themselves, activates their WM.

P7: *"The intention is that, first, you help them with it, and then, when they articulate it, they start to grasp that tactic. So, you do support them, but you also let them know that*

*'We've said it, we're doing it step by step, so think along'. I find it very important that they realise themselves what they need to bring along."*

***Adjusting support (individualised approach).*** These strategies are part of 'het zorgcontinuüm'—a support model that aims to provide appropriate care and guidance to students with specific educational needs. The model consists of four phases that follow each other: broad basic care, increased care, extension of care, and an individually adapted curriculum. Broad basic care is offered to every student. However, when that does not suffice, a teacher should provide increased care. In this phase, additional measures: remediation, compensation, differentiation, stimulation, and dispensation, are taken at the school and classroom level to ensure that the student can continue to follow the common curriculum. While the model primarily applies to the Belgian education system, the Netherlands has a similar system in place to support students with special educational needs and to provide appropriate care and guidance. All interviewed teachers mentioned adapting their support to individual needs, thus highlighting the importance of these strategies.

Remediation. Remediation refers to the additional support that students who are falling behind (i.e., including those with WM difficulties) receive. All teachers employ various methods to remediate based on the student's specific needs and difficulties. For instance, they give extended instructions to the children with poorer WM to further explain the lesson content while other students are already working independently.

P1: *"A lot of children, especially children with special needs...I take them aside. We already have a number of children where you say, 'Okay, we've had that structure', for example, giving extended instruction in small groups...then you see that there are a lot who really make a big leap in two months' time, including in WM."*

Other ways to help students initiate work on their assignment include: opening their workbook, marking the exercises they have to make, covering irrelevant exercises with a cover sheet, and turning the page when they are ready.

Compensation. Fifteen teachers indicated compensation as a strategy to support children with WM problems. It involves implementing alternative methods or techniques that allow students to overcome their WM-related difficulties. Teachers mentioned using materials (e.g., post-its, pictograms) to help students with WM difficulties stay organised and focused.

P3: *"What I do additionally for those children is, if they have difficulty remembering things briefly and you really know that it's a problem, then we work with help cards or a help folder. So they can look up information there, or for other children, who make letter substitutions, I stick a card on the desk."*

Differentiation. Nine teachers mentioned differentiating instructions to support children with WM problems better. A few teachers mentioned using a 'three-track' approach, which involves dividing students into smaller subgroups based on their instructional needs. Due to this, the teacher can allocate more time and attention to the group of children with WM problems.

P10: *"That is also because we work with tracks. So at the back, there are blocks where the children usually start working after a short instruction. And then I have my two rows at the front. You can think of the row right at the front, where I constantly go by or sit next to...with my table that slides around because I always have to intervene. Because they no longer know what to do."*

Stimulation. Seven teachers mentioned stimulation as a way to support WM. In this context, stimulation refers to the teacher actively encouraging a student. It can take various forms, such as using songs, games, giving feedback, or prompting. Such strategies can make the content more attractive and entertaining, and keep children engaged and dedicated to the task, even after having faced a difficulty.

P5: *"Yes, even though they don't have the answer at all, the fact that you say, 'Oops', or show any kind of doubt, those children lose their confidence. But by responding positively*

*and saying, 'I see that you're thinking very well, do you remember what we saw there? Who can give a tip here in the class?'. Then ask, 'Can you take it from here?', and then say, 'I'll come back to you later, alright, but let's hear what the others have to say'."*

Dispensation. Only two teachers mentioned dispensation as a strategy to support students with WM problems. Dispensation means exempting students from specific assignments or parts of the exercise so as to not overload their WM. For instance, students may have to complete fewer exercises or homework tasks.

P1: *"When you're younger, and you're even younger in your head, because. . .not everyone has the same maturity at the same age, when I see that, when I realise that, then I think 'I need to either dispense something and drop something, or compensate by giving something else, because it's just a bit too much'."*

*Monitoring.* Teachers dedicate a substantial amount of time to closely monitoring the student's progress and ensuring the understanding and remembrance of instructions. This is especially important for students with WM problems, as indicated by twelve teachers. Keeping a watchful eye on students can be seen as a proactive measure to prevent them from getting lost or falling behind, as outlined in the teacher-provided example. Furthermore, it can be a way to regain the students' attention and to have them reflect on what to do next.

P3: *"Then I try to. . .make sure to frequently go back to those children. Pay attention to the others as well, but instead of checking on that child every 10 minutes, make sure to go and see them at least every 5 minutes. Because otherwise, for example, within half an hour, that child may have completed 5 exercises, and all 5 of them could be wrong."*

*Critical thinking.* Critical thinking is the ability to engage in reflection, analyse information, and effectively solve problems. This was highlighted by thirteen teachers. Teachers actively encourage children to seek answers independently, which also contributes to using and refining their WM, as discussed before. Teachers do this by asking (follow-up) questions like 'What helped you to figure this out?', 'How can we do it better next time?', 'Why do you think it is important to learn about this?', or 'What is your neighbour/the class doing?'. Encouraging children with WM difficulties to actively reflect on their actions can help them better understand the situation, analyse and evaluate it, and develop alternatives/solutions, in turn, increasing their metacognitive knowledge to better use the limited resources and, thus, improving WM.

P5: *"Constantly engaging in a conversation about their own thinking. 'What has helped you now? Because I can see that you understand it. Let's write that down here so that next time, if you don't remember, we can review it again'. And then tomorrow, when we do it again, we'll think about that. 'Do you remember yesterday when we thought about it really hard? But we knew what helped you back then. Do you remember that?'."*

Teachers indicated engaging their students' critical thinking by encouraging them to use the available resources (e.g., school diary). A few teachers mentioned that children often can come up with suggestions regarding how to improve their assignments or how to seek and receive help from their peers.

*Prompts and cues.* By prompting, teachers give subtle hints or suggestions to aid students experiencing WM difficulties in their learning process and reinforce their memory retention, as highlighted by eight teachers. Teachers implement this strategy by repeating information in unison with the children. This synchronous repetition helps solidify the knowledge or concept in the students' minds, otherwise struggling with recall. Another approach is to have students initiate or complete the instruction themselves, thereby actively engaging their WM and involving them in the learning process. Furthermore, some teachers use contextual reminders to enhance student understanding.

P9: *"I say it like, I don't know, loudly, or I say it very exaggeratedly, very foolishly like that. So that afterwards, they say on the test, 'Oh yes, decimal numbers are not allowed because that's what the teacher was being so foolish about'."*

Regarding students with WM difficulties, teachers prompt the children before the problematic behaviour occurs by incorporating visual aids, such as pictograms or pictures, to supplement the instructional materials. Pointing at the relevant visual stimuli or the instruction/overview (presented on the blackboard or attached to the student's desk) or providing a verbal prompt of the next step (before/if the child gets stuck during a task) can facilitate children's learning. By having visual cues available, children with WM difficulties can refer to them whenever necessary.

*Classroom arrangement.* Classroom arrangement refers to the structuring of the physical environment to enhance student learning by reducing unnecessary distractions. Among the interviewed, thirteen teachers identified classroom arrangement as valuable when supporting children with WM difficulties. One of the classroom arrangement strategies involves avoiding distracting stimuli while completing assignments.

> P1: *"That should also be somewhere on their desk, I say, 'You put it there, and you only take what you need for those exercises'. It's a very clear instruction, so they're really not occupied with other things, not even in their minds. They're not thinking about flowers or... No, it's really about 'I use that equipment to do my exercises'."*

Six other teachers mentioned the organisation of materials (e.g., having designated places). It helps to create a structured learning environment where students can easily locate and access the necessary materials, promoting efficiency and focus during classroom activities. Another effective strategy for classroom organisation involves arranging the physical classroom environment (e.g., seating arrangement). By strategically arranging the desks, teachers create an environment that optimally supports the needs of students, particularly those requiring additional assistance.

> P4: *"I have a U-shaped arrangement of desks at the front of my classroom, with 6 children sitting around me, 3 on each side, so I can sit between them. This is what I call the instruction corner. If I notice that a child at the back of the class is struggling, I ask them, 'Would you like to come sit with the teacher for some extra explanation?'. Most of the time, they say yes. They can come to sit around me, and then I have those children very close to me, allowing me to go through everything step by step again."*

As mentioned by one teacher, the classroom can be subdivided into groups/corners or even separated into different rooms.

> P4: *"If there is noise. . .that's an additional stimulus. They already have to manage the task, the overview, the hour, keeping everything in mind, and then there's the noise on top of that. That's why we chose those two classes. A quiet class and a chatty classroom. And in the chatty classroom, they can work in pairs. It's just a buzz, but sometimes it can be disruptive."*

*Varying modalities/materials.* Teachers employ a variety of modalities and materials that help children with WM-related difficulties by reducing the demands on their WM and providing them with multiple ways to process and remember information. Sixteen teachers emphasise the effectiveness of visual stimuli, such as pictograms, instructions on the blackboard, counting blocks, and videos. Others utilise auditory stimuli, including songs, poems, and the repetition of instructions. One teacher also mentioned the use of 'corner work' and games. These methods are often implemented at the classroom level to benefit all students and can aid students' engagement with the lesson content. However, teachers also recognise the importance of an individualised approach when addressing the specific needs of children with WM difficulties. By using multiple methods and visual and auditory aids, teachers can relieve the demands placed on children's WM.

> P4: *"I didn't write it down and just said it. But then there are children who are not auditory learners. Their memory cannot keep up. And I'm already three steps ahead while they're still on step 1 and haven't caught up. So visualising it helps. . .because earlier, when we were working on it, the visual aspect wasn't sufficient. So I think it needs to be presented in two ways."*

3.1.2. Self-Determination Theory-Based Strategies

The literature on self-determination theory emphasises that students are motivated by activities meeting their needs for competence, autonomy, and relatedness, with certain instructional practices bolstering these needs and improving learning outcomes.

**Teacher's rationale.**

Even though there was no explicit mention of this theory in the teachers' responses during the interviews, the outlined strategies seem to correspond with the principles of self-determination theory; more specifically, the basic psychological needs (competence, autonomy, and relatedness) and motivation spectrum (from extrinsic to intrinsic motivation).

**Strategies.**

*Promotion of student interest.* Teachers play a crucial role in supporting children's WM by employing strategies that enhance attention, focus, and active listening, as mentioned by eleven teachers. One approach mentioned by the teachers was to accompany verbal instructions with expressive gestures. By articulating instructions clearly and using animated gestures, teachers capture students' attention and promote active listening. Teachers further mentioned incorporating brief physical activities, such as stretching or deep breathing, to help children (re)focus.

P10: *"They help to relax for a moment, after, for example, a very difficult class. But they also help to refocus on the next one, actually."*

Four teachers further emphasised the need for a quiet classroom environment (during instruction) to promote and maintain student interest and attention.

P8: *"What we actually started with, is to make it very quiet when an instruction is given. Silence is a requirement to be able to listen to an instruction…if it's not quiet, it's normal that children can't focus anymore."*

For students with WM challenges, noise-cancelling headsets can be a proactive solution whenever the classroom is noisy. Such headsets help minimise auditory distractions, enabling students to maintain focus during independent work.

*Providing feedback.* Feedback can be defined as providing information to students regarding their learning process. It involves sharing evaluations or suggestions that present students with insights into their performance and progress. Fourteen teachers confirmed that they frequently provided some form of feedback to their students experiencing WM difficulties. The feedback, most commonly, is centred on task performance and the underlying process. With the help of feedback, the teacher indicates whether the task was successfully completed, highlights areas for improvement, and examines and reflects back on how the student approached and executed the task. When presented in a constructive manner, feedback can enhance students' awareness of their strengths and sense of competence. By providing feedback, teachers help students reflect on their learning process and identify areas where they need additional support, thereby allowing them to recognise their weaknesses and seek help, thus fostering autonomy. This can promote metacognitive awareness and help students develop their WM capacity. Feedback at the task and process levels is often combined with feedback on the self-level, where teachers specifically focus on the qualities of the students. One teacher provided an example of supporting students who experience WM-related difficulties with a task by focusing on their strengths and overall effort.

P2: *"I also try to support children who find it more challenging, for example, by saying, 'I see that you're really good at addition, but subtraction is still a bit difficult, isn't it?'. Or when it comes to long division, I might say, 'You've done a great job overall, but there are some errors in the division exercises. Let's take a look together at what went wrong there'. It's always about balancing what is going well and what we still need to work on."*

*Autonomy, leadership, and responsibility.* Sixteen teachers emphasised the importance of fostering students' capacity to work independently and encouraging them to

take responsibility and charge of their own learning process. According to the teachers, autonomy and taking responsibility are important for students to develop and utilise their WM and acquire other non-WM-related skills.

Several teachers also noted that providing choices can be an effective means to support students' autonomy. One teacher explained how she implements a 'three-track' approach to promote students' independence and sense of autonomy by allowing them to choose their group.

> P5: "*Children who are struggling, come to my mini-classroom [blue track] and receive the basic programme. Then you have the green track for students that can handle the basic programme independently, without the teacher. So, alone. And then you have the red track, which is the extension. These are children who say, 'I can do this, I'll do it alone, but give me a bit of a challenge'. By visually presenting it and letting the children determine which group they feel they belong to, it allows them to take their learning process into their own hands.*"

A couple of teachers discussed offering the students choices in selecting tasks, materials, and strategies. Offering choices has the potential to positively influence students' intrinsic motivation and enhance their engagement in the learning process.

> P6: "*What I do, in this class, is work on intrinsic motivation. Actually teaching them why it is important to always do that. Because, as teachers, we are used to just taking care of it. But let them think for themselves, 'Why is that?'.*"

Moreover, taking responsibility included instances where students assumed responsibility for their own materials (e.g., taking home the books needed), completing and submitting assignments on time, and through actions, such as assisting their peers by acting as a buddy.

***Student expression.*** The value of enabling students to voice their own opinions, specifically concerning their learning environment, lesson content, and personal issues, was emphasised by seven teachers. Regarding children with WM difficulties, instead of simply dictating instructions, many teachers recognise the importance of collaborating with their students by actively seeking student input and listening to their opinions and preferences. For example, this might involve discussing whether a student feels comfortable sitting in front of the class.

> P6: "*What I find important is, for example, a child who frequently forgets their belongings. In such cases, I think it is important to find out the reason behind it. I'm not a big fan of saying, 'If you forget your things five times, there will be a consequence'. Sometimes, there are trivial things that families expect, which may explain forgetfulness. Just listening to the children and giving them the opportunity to explain themselves is also important.*"

***Providing an overview.*** Providing an overview of each day's schedule and indicating specific times for different activities/tasks can further support children with WM difficulties. Given that the schedule is displayed on the blackboard, it provides the students with a clear understanding of the agenda and allows them to anticipate what will happen next. This is especially helpful for children with WM difficulties, as it enables students to efficiently manage their time, plan ahead, and (mentally) prepare for the upcoming activity.

> P10: "*There are children where I could say in the morning, you do this and that. And that I write on the board when I'm going to give instructions about what. I let them start anyway. I say, look, this is your schedule, but I write on the board that at 10 o'clock I will explain how to measure angles. At 11 o'clock, I will explain sentence analysis. So they know, okay, we can do all those other things, and then we'll get instruction on that.*"

### 3.1.3. Social Learning Theory-Based Strategies

Based on the literature, social learning theory posits that in a structured classroom, students internalise behaviours through observation, eventually reducing the teacher's role as an external regulator and promoting autonomous learning.

**Teacher's rationale.**

While teachers did not explicitly reference this theory during the interviews, the literature suggests that the strategies described are grounded in social learning theory principles, such as observational learning, retention (memory), and reproduction of the behaviour.

**Strategies.**

*Routines.* According to eight teachers, routines are essential for children with WM problems as they provide predictability, consistency, and structure. Consistency allows children with WM-related difficulties to rely on familiar patterns and cues. Teachers have specific routines at the start or end of the day/lesson, including the organisation of materials and activities (e.g., collecting homework and emptying the school bag).

P5: *"But due to the fact that I alphabetically pick up my homework every morning, they know it."*

*Rules.* Establishing rules is a valuable strategy teachers employ to support children in remembering important information. According to nine of the teachers interviewed, rules serve as effective reminders for children, reinforcing what they need to do. Moreover, rules are considered non-negotiable, leaving no room for debate or confusion. This strategy is particularly beneficial for children with poor WM, who struggle with memory retention and cognitive processing, as it provides clear boundaries and expectations, reducing the cognitive effort needed to remember specific expectations.

P6: *"In the reading corner, there are rules. They are not displayed there, but they are simply things that should logically be established."*

Teachers use various methods to help children remember rules, utilising visual and verbal stimuli. As emphasised by one of the teachers, they may utilise visual aids such as pictograms or employ verbal reinforcement by repeating the rules.

P8: *"We are going to learn that there should be no walking in and out. No more 'Oh, I forgot something, let me go get it'. So we have put up a pictogram saying that once you are outside, you cannot re-enter for something you forgot."*

*Behaviour management.* External behavioural management can be understood as a proactive approach employed by teachers to establish clear expectations, consequences, and redirection strategies to manage student behaviour effectively. The importance of external behavioural management in supporting children with WM problems was recognised by sixteen of the interviewed teachers. Among these teachers, establishing clear expectations emerged as the most important aspect of external behavioural management for students with WM difficulties. One of the teachers illustrated this by providing an example of using a timer to clearly communicate expectations to the students (as both an individual approach and a whole classroom approach). The importance of time management and providing reminders (of remaining time) was highlighted by several teachers as an effective strategy.

P14: *"Yes, we have, for example, two children in the class who have a timer. I can set it for them and say, 'You will now work for five minutes, solving problems. And when it goes off, you'll do this'. Sometimes, I also do this for the whole class. If they all need to do something, I say, 'Now, all of you will first do this, and when you're done, you can do that'. Only then, I give the third task. It's more in blocks, no longer a whole hour. But first, you do this, and then that."*

Furthermore, teachers recognise the value of letting children face challenges and experience frustration. Such an approach provides students with opportunities to observe and imitate problem-solving strategies used by others, resulting in an improvement of their problem-solving abilities and the activation of their WM.

P13: *"But then they start cooking, and because they have to train here, I also prepare as little as possible because the task here is for them to face something challenging and learn by themselves, and to receive guidance in that."*

Moreover, eight teachers use rewards as effective tools for supporting children with memory retention difficulties.

P18: *"But we also have the agreement that if he gets twelve checkmarks every week, then he can choose a Just Dance on Fridays. Then the whole class dances to whatever he wants. It's that simple."*

Finally, the redirection of misbehaviour is another important aspect of external behavioural management, as highlighted by nine teachers. Through redirection, teachers guide students with WM-related challenges to reflect on their actions and adapt and redirect their focus appropriately.

P2: *"I know my students who start working too quickly and end up doing it in a different way than what is asked. Instead of saying, 'Hey, didn't you notice you were supposed to...', I say, 'Stop working, I'll take a look over there. You read the instructions, and after, you'll tell me what you actually need to do'."*

*Modelling.* Only three teachers spoke about modelling as a way to support students with WM problems. These teachers show what the students have to do and have them imitate this behaviour.

P5: *"I am a bit like her memory, so to speak. And then it is indeed showing how it's done."*

One teacher also emphasised that students often copy each other's behaviour, hinting towards peer-led modelling, which can be especially effective for supporting children's WM.

P3: *"How we end our day is by bringing in our school bags and preparing them together. I attach pictograms to the board indicating what needs to be packed. I demonstrate it to one child, and then they all have to prepare the items and place them on their desks before putting them in their school bags. Then they have to come and collect everything before their parents come."*

*Self- and parallel talk.* Children with WM problems can benefit from the teacher mapping his/her own or the students' actions through language and description in order to better comprehend the steps/actions that need to be taken. Repetition was considered one of the most beneficial strategies teachers employ with students with WM problems, leading to better retention.

P9: *"I make them repeat what I just said often. And I often point out those who will not remember it. If I have given three instructions, then I ask, 'What did I just say? Repeat it again'. Then I have each thing repeated separately by someone. They really need to know, 'I have to say what we have done, then I need to know what the others have already said'. They really need to listen carefully."*

### 3.1.4. Attachment Theory-Based Strategies

Drawing from the literature on attachment theory, teachers' emotional support fosters a secure learning environment, building trusting relationships, enhancing students' emotional well-being and, consequently, positively influencing their WM.

**Teacher's Rationale.**

*Safe haven.* By providing emotional support in the classrooms, teachers assume the role of a safe haven for their students. As a result, students can develop a sense of trust in their teachers, feel at ease when sharing their thoughts and concerns, and seek guidance when needed. Four teachers expressed that being a safe haven for children is fundamental for the effectiveness of the strategies employed when dealing with WM-related challenges. When students feel safe and supported by their teachers, they experience lower stress and anxiety levels. Such emotional regulation creates an optimal environment for WM functioning, allowing students to allocate cognitive resources effectively to the task. Being

emotionally supportive, therefore, becomes a prerequisite for creating an optimal learning environment, as explained by one of the teachers.

P7: *"If a child doesn't feel comfortable with you, no matter what you try, as a teacher, you're stuck to some extent because then you can ask and say whatever you want, but it's stuck."*

Moreover, when a child does not feel emotionally secure, their WM can, in turn, be negatively affected. Emotional distress might overwhelm and occupy a child's mind, making it challenging to engage with tasks and employ their WM effectively.

P4: *"I think that if those children don't feel emotionally safe, nothing else can succeed either. So, WM won't work, either. Because in my head, that child's mind gets so full when they don't feel well that everything else doesn't want to come along, and there are too many other stimuli in the head."*

*Secure base.* In addition to being a safe haven, six teachers expressed that providing a secure base can effectively aid children's learning. Students who feel safe and supported are more likely to explore, ask questions and actively engage in learning.

P1: *"In my opinion, the first step, when a child feels safe, and when they feel supported, and when a child feels 'I can say anything here', so to speak, then that's already a very good start, then there is that peace."*

Furthermore, noticing and appreciating students' active engagement can positively impact the student's confidence. As highlighted by the teachers, the student can feel empowered and, in turn, continue his/her efforts in learning.

P18: *"You also need someone who can say, 'Have you thought about this or have you thought about that?'. Yes, things she has to think about herself. Sometimes she needs help with that, and that gives her the peace of mind to be able to carry it out."*

Using such teaching strategies to support students with WM problems might provide them with a sense of competence. As outlined by the teachers, by offering reassurance, teachers help students to feel competent and capable of carrying out tasks and overcoming challenges related to WM difficulties.

P13: *"That you actually can have control over yourself. And over your own actions. That, yes, you can have a grip on that."*

**Strategies.**

*Emotional safety.* According to eleven teachers, providing emotional safety can be defined as creating a supportive and nurturing environment where students feel secure, accepted, and free from judgement. When considering children with WM difficulties, five teachers stressed that the classroom is a place for learning and highlighted that making mistakes is normal and a part of the process; this can be especially comforting for children with WM-related challenges when they 'get stuck'.

P2: *"I mainly try to make them feel that I am here to help you. Giving them a sense of safety. Making them feel that it's okay that you can't do it. You come to school to learn it. If you could already do it, what would be the point?"*

Three teachers mentioned that they avoid belittling children in front of the class or putting them in an awkward position whenever they are experiencing trouble.

P10: *"I will pair a stronger one with a weaker one. Because I want them to experience success, and I know that if I put those two weaker ones together, they will definitely get stuck. I also don't want to embarrass them."*

Additionally, seven teachers highlighted the importance of remaining calm and patient whenever a child forgets what to do. These teachers preferred to directly engage with the child with WM difficulty, openly address the problems or challenges, and recognise their strengths and individual successes. When children feel safe to express themselves and

communicate openly and adequately, they are more likely to develop respectful attitudes and behaviours towards each other, thereby contributing to a positive classroom climate, as highlighted by a couple of teachers.

> P6: "*Sure, everything should be able to be said here. Not everything to the whole class, but even if something doesn't work out, if they don't understand something, raising their hand, it all has to do with such a safe living environment. That everyone feels like what succeeds is good, what doesn't succeed is fine. This is a school for learning. And not just learning arithmetic, but learning who am I and what are my strengths and weaknesses.*"

*Attuned communication.* Four teachers highlighted that creating a supportive classroom environment through positive communication is crucial, thus minimising negative communication whenever possible. Regarding WM problems, two teachers indicated that they try to limit negative communication on an individual level by tapping the child's desk or pointing at a visual stimulus instead of approaching the students when they are showing undesirable behaviour.

According to a couple of teachers, using humour aids positive communication and, in turn, a positive classroom climate.

> P7: "*But in a childish way, okay. We take this book, lift it up in the air, and now put it in the school bag. And then they'll laugh their heads off, but they'll also know 'uh-oh'. And you'll see those who always forget it, like, 'Yeah, I know it'.*"

On the classroom level, praising students, especially those struggling, is another important aspect of positive communication, as highlighted by two teachers. Publicly praising students for their achievements creates a supportive classroom climate where students feel recognised, valued, and motivated, which is particularly important for empowering children with WM difficulties.

> P2: "*You also notice that children are more engaged. And when children are more engaged, they are thinking about it more, and you can give them a compliment afterwards. And when you give them a compliment, they will start doing it themselves in the future. Giving compliments and praise is very important, you know. I always correct discretely, but I always give praise in a big way. So that everyone hears it, and it becomes contagious for others.*"

*Responsiveness.* Thirteen teachers highlighted the importance of being responsive in one-on-one relationships with students facing WM difficulties. In the current context, responsiveness refers to teachers being easily approachable or emotionally available and encouraging honest and open communication about their own and students' emotions. Six teachers highlighted that being emotionally available and understanding can provide insights into a child's specific (WM-related) difficulties and allow the teacher to provide appropriate and tailored support.

> P7: "*You had to first figure out what's wrong. Is it because he doesn't like coming to school? But that wasn't the problem; it was the home situation. And then address that by talking to him, saying, 'You can always come here, but it's not an obligation'. And that's when he started talking about it. And sometimes I would say, 'Let's sit down separately and talk about it'.*"

Furthermore, teachers also emphasised the importance of addressing and showing understanding towards students' WM problems and concerns by actively inquiring about the problem while still being respectful.

> P9: "*And I asked her earlier how things are because 'I see that there's a lot going on and that you're so uncertain'. And she started crying immediately. It's often a personal conversation in which they say if everything is okay. That it comes out right away. It could also be that you have to dig a little deeper or try something else.*"

Five teachers focused on encouraging and affirming the student when facing WM-related difficulties. A few teachers also talked about the goals they want to achieve when be-

ing responsive towards students, namely to give the student confidence and to let him/her experience success or to lower pressure and stress in the classroom. Teachers particularly emphasised the importance of this for children who struggle with task organisation and completion—a common difficulty among those with poor WM.

> P5: *"Children who perceive themselves as anxious about failure sometimes need to be allowed to do so and experience moments of success. And then challenge them a bit. And say, 'You can do this, let's try a little more. And it's okay if it doesn't work out; I'm here for you. But I believe in you, that you can do it'."*

Additionally, being responsive includes acknowledging students' efforts and accomplishments, both in terms of the process and the outcome, which is particularly important for children who struggle with task organisation and completion.

> P2: *"I also try to support children who are struggling, for example, saying, 'I see that you're really good at addition, but subtraction is still a bit challenging, right?'. Or with long division, 'You did a great job overall, but there were some mistakes in the division exercises. Let's take a look together at where you went wrong'. It's always about acknowledging what is going well and identifying areas where we still need to work on."*

Moreover, two teachers acknowledged that a positive attitude when dealing with children with WM difficulties can effectively counteract students' frustration and help them face and address challenges.

> P7: *"I want children who are struggling with something to be able to approach it in a way that goes like, 'Well, it's just something I'm not good at right now, maybe I should just laugh about it'. That's looking at things in a positive way. You can't be good at everything."*

*Sensitivity.* Based on the interviews, teacher sensitivity can be seen as a teacher's awareness of students' individual needs, differences and support-seeking behaviour, and noticing distress and problems. A total of thirteen teachers highlighted the importance of being sensitive, while six teachers emphasised being aware of children's differences and the need for an individualised approach when providing support (i.e., adapting expectations and using appropriate strategies), particularly applicable to students experiencing WM-related challenges.

> P14: *"It's actually on an individual basis that you look at what do you need and why are you coming with this request for help. Is it a lack of skills, or is there something else behind it? And then I adjust the strategy accordingly."*

Furthermore, as indicated by four teachers, individual preferences are evident when seeking support. While some students with WM difficulties may proactively approach the teacher for assistance, others require the teacher to take the initiative and offer support. This implies that teachers need to show flexibility and adaptability in their approach. Moreover, the teachers' ability to perceive signs of distress in students is closely tied to the concept of awareness, as highlighted by eleven teachers. One teacher reflected on her ability to identify underlying emotional issues and adjust her strategies to provide adequate support.

> P8: *"It's always a bit of a search to understand why a child behaves like that. Because I had a child here who hardly ever completed anything and didn't care about homework and such. But then it turned out that there were emotional issues going on at home. And I do notice that when you pay attention to that, and you have a chat with the child, things improve. And sometimes you can even tell the child, 'Okay, let's reduce or skip the homework for now'."*

### 3.1.5. Metacognition

**Teacher's rationale.**

Five teachers emphasised the importance of encouraging children to think independently and reflect on their own learning to foster autonomy and self-reliance. This per-

spective resonates with self-determination theory, which links autonomy to increased motivation and engagement. Additionally, by promoting self-reflection, teachers support the development of metacognitive skills, aligning with the principles of social learning theory. Some interviewed teachers also highlighted that aiding students in recognising their own forgetfulness can enhance metacognitive skills. Drawing from cognitive load theory [102], it is evident that by providing strategies to counter forgetfulness, teachers mitigate cognitive overload, further bolstering learning outcomes, including WM.

> P9: *"If they have discovered it themselves, they will remember it much better afterwards."*

By being mindful of their own forgetfulness, children are better equipped to develop creative solutions and strategies to prevent forgetting important information in the future.

> P6: *"You teach them to reflect on everything they do, and I think that's also very important for improving WM or figuring out 'How does that work for me?'. And when you encounter certain problems, that's also the only way to know, 'Okay, something needs to be done about this'."*

3.1.6. Attention, Focus, and Motivation

**Teacher's rationale.**

Teachers highlighted the importance of children's attention, focus, and motivation in their teaching strategies supporting WM. These emphases from the teachers resonate with the principles of cognitive load theory, aiming to reduce the unnecessary cognitive load and enhance the relevant cognitive load to facilitate learning. Additionally, the emphasis on enhancing motivation seems to draw parallels with self-determination theory, which encourages the promotion of active engagement and ownership in the learning process.

*Attention and focus.* Multiple teachers believed that strategies focusing on supporting WM problems are effective because these strategies engage and maintain children's attention and focus, consequently leading to an enhanced concentration on tasks. Teachers highlighted how distractions hinder children's engagement with the task.

> P10: *"And then they just don't get to it. They simply don't even manage to listen to what the assignment is supposed to be. Because that also requires concentration."*

As teachers provide emotional support (e.g., establishing eye contact when talking with the child) and classroom organisation (e.g., informing the child about expectations and consequences of certain behaviour), they are able to enhance the attention (and WM) of their students.

> P8: *"That children simply don't pay attention anymore. Even children who should be able to do so just can't remember because they are actually not focused anymore."*

*(Intrinsic) motivation.* Motivation serves as a fundamental rationale underlying the effectiveness of teacher strategies for children with WM problems. While attention and focus play a crucial role in children's ability to grasp and comprehend an instruction or task, these factors alone do not determine students' success. In order to improve students' engagement with a task, it is equally important to cultivate their intrinsic motivation, as highlighted by three teachers. A lack of interest and motivation in an activity/task can hinder students' attention, resulting in their WM remaining inactive.

> P5: *"Because it's quite a passive situation when you're just sitting in class. And then you have to activate all of that if you're not motivated, and it doesn't work well."*

Nevertheless, teachers stressed that by enhancing students' intrinsic motivation, they could promote the effective engagement of their WM. When students find a task interesting and engaging, they are less likely to forget relevant information, thereby activating and improving their WM.

> P17: *"I think that regardless of everything, if you see the usefulness in something, you will then forget it less quickly."*

Motivation can be increased by providing children with meaningful choices. Giving students the opportunity to make decisions about their learning, in turn, allows them to feel empowered and engaged.

> P5: "*But by visualising it, and by letting the children themselves determine which group they think they belong to, it also enables them to take control of their learning process, which makes them feel good about it and motivated. The fact that you genuinely have it in your own hands gives freedom and a sense of security and relaxation in which everything works better.*"

Such a sense of autonomy enhances students' intrinsic motivation and contributes to a positive classroom atmosphere.

> P6: "*And after a while, they realise that they feel better themselves, that the classroom atmosphere is more pleasant when things go smoothly.*"

Finally, extrinsic motivation, such as wishing to receive praise on the one hand, or not wanting to disappoint people or experience frustration on the other hand, can serve as a way through which the strategies work, as further highlighted by a couple of teachers.

> P9: "*They know they can be asked to repeat it and they're afraid of not being able to answer or getting a remark. That can be simple, you know. So they listen a bit more attentively when I ask, 'What did I just say?'. Because I also notice that they feel embarrassed when they can't answer. And they think, 'Oops, it wasn't good of me not to listen now'.*"

### 3.1.7. Repetition and Routine

**Teacher's rationale.**

From the perspective of seventeen teachers, the importance of incorporating repetition and establishing routine in the classroom to support children's WM was emphasised. This approach aligns well with the principles of cognitive load theory and, to a certain extent, the tenets of social learning theory.

Repetition can make children less prone to forgetting instructions. When instructions are repeated, children have more opportunities to absorb the content, leading to the repeated activation of their WM and stronger retention.

> P12: "*I do notice that when you keep asking the same thing to children, it does help. If you consistently repeat those same steps.*"

Additionally, repetition helps reduce the cognitive load on children's WM as the instructions become routine (through automatisation).

> P6: "*If that basic structure isn't there, you will always end up losing children who are really blocked by that, who can't function without that organisation.*"

### *3.2. Influencing Characteristics and Factors*
### 3.2.1. Child Characteristics

Teachers frequently mentioned the specific character traits of the child as an influencing factor. Adapting the teachers' support and approach to the specific needs of their students is essential for improving the child's outcomes (including WM).

> P10: "*And then you also have the difference between children who accept help more easily and those who accept it less easily. You'll notice that sometimes you have to take a step back. You don't have to be there all the time because they won't be able to function properly then.*"

Teachers identified several other child characteristics impacting the effectiveness of the strategies employed. These traits included the children's personality (e.g., introverted versus extroverted), temperament (e.g., dominant versus humble), cognitive development (e.g., dependent versus independent), social development (e.g., playful versus responsible), motivation, and basic levels of attention and concentration. Teachers recognised

these factors as influential aspects of the students' engagement with and response to the employed strategies.

Additionally, teachers noted that children differ in their preferences, especially regarding information processing. Some children prefer or benefit more from auditive instructions, while other children prefer to receive visual instructions.

> P2: *"In regards to visuals, I simply notice differences among children. There are children who understand it better when I explain it, and there are other children who really need that visual. Even if I repeat it 10 times, they still can't repeat what I said."*

Age was another important factor mentioned as influencing the effectiveness of the strategies, particularly in those with WM problems. Younger children often encounter more challenges when working independently, while older children tend to show greater autonomy in their learning process. One teacher noted that younger children often rely on imitation as a means of remembering information, whereas older children might use other strategies.

Furthermore, some teachers mentioned that whether children are showing symptoms of attention deficit-hyperactivity disorder (ADHD) or autism spectrum disorder (ASD) also influences the strategies they use. For these children, clear and simple instructions are preferred over elaborate explanations.

> P3: *"Especially with children with autism, very brief and clear. Just a straightforward instruction. And that's definitely a tip that I really use, especially for those children. Not using more words, no elaborate explanation about why they should sit, for example. Just stop, sit, be quiet."*

### 3.2.2. Teacher Characteristics

Several teachers have mentioned specific teacher characteristics that can impact the effectiveness of strategies to support students. Three teachers specifically emphasised how their own organisational skills can influence their strategy use. Being structured and organised can serve as a starting point and an effective example. On the other hand, a chaotic approach may hinder the effectiveness of classroom organisation and the establishment of a structured daily routine.

> P1: *"It's important that you know in advance what you're going to do as a teacher. Because if you think that you're just going to do something here, well then, it also won't work because then you don't have clarity in your own mind and you can't communicate it clearly either."*

Another important teacher characteristic is the (years of) experience supporting children with (WM) difficulties. Having more experience equips teachers with a better understanding of the challenges these students face and helps teachers to refine their strategies further. However, as one teacher pointed out, even with extensive experience, there is always a possibility for new challenges to arise.

### 3.2.3. Learning Environment

**Peers.** Peers play an important role in the learning environment as they, aside from the teacher, can be seen as a valuable source of support. Interacting with their peers allows students with WM difficulties to observe how they can organise information and tackle assignments. This way, peers can actively contribute and facilitate the teacher's implementation of (instructional) strategies.

> P10: *"And then there are already a few students who pick up on that; we had to do this and this. So the students also often help you with that. And there are also a few who set a good example, and that also helps."*

Furthermore, teachers can group students with WM difficulties with those who have stronger WM skills. An important benefit of such grouping is that peers can actively assist students in remembering information, thereby supporting their WM. However, as one

teacher indicated, there can be a potential downside to peer interaction—competitive behaviour. A competitive environment can pose further challenges for those with difficulties and somewhat hinder the creation of a supportive and respectful learning climate.

**Classroom context and processes.** In the classroom, various factors can influence the strategies used and their effectiveness. Based on the interviews, a distinction can be made between classroom context (organisation) and classroom processes (characteristics of the strategies).

Firstly, the influential role of the classroom context was discussed by two teachers. One teacher pointed out the potential drawbacks of having an alternative classroom setup (e.g., a 'three-track' approach). Such a setup may result in some students losing visibility of the blackboard and instructional materials. The teacher should consider this when assigning children with WM difficulties a seat. Another teacher, on the other hand, expressed challenges met when providing individualised support. Distractions caused by excessive noise and movement by other students can overstimulate the students and hinder their ability to concentrate. Ultimately, it can impact the student's capacity to understand, remember, apply the instructions provided, and engage with the material.

Secondly, five teachers also discussed the specific characteristics of the strategies they employ. These included factors such as the duration of the strategy implementation, the variety and alternation of strategies, and the level at which the strategy is applied (classroom or dyadic level).

> P8: *"But classroom instruction has become very difficult for children. No matter how short it is, it only gets harder and harder. Sometimes, I have the impression that when they can just work on a sheet and not have too much explanation, they are better off. And yes, variety, if you have a lot of variety, they are also more engaged. It shouldn't all last too long or become too boring."*

Furthermore, another teacher emphasised the importance of shifting and flexibility in teaching approaches (e.g., corner versus contract work) and providing children with choices (e.g., starting and completing an assignment immediately or later). However, this approach can pose challenges for children with WM problems, as they may struggle to recall instructions accurately, given the lack of precision and structure.

> P2: *"They receive various assignments together, which they can start immediately or start later. Often, the instructions are given in the morning, but they only begin the task after recess. So there is a considerable time gap in between. When you work like this, naturally, you have those children who are weak in terms of that WM, as you call it, who struggle with it. They forget, saying, 'Yes, the teacher explained that this morning, but I started with math first. Teacher, how was it again? How did you say that?'. And then you notice those children who have a strong WM, it doesn't affect them, they perform consistently throughout."*

**School context.** The wider school environment further influences the strategies employed by the teachers. Several factors within the educational system can impact the approach taken by the teachers. One teacher highlighted the pressure she experiences in measuring up to the expectations of the educational system. High academic standards can create additional stress for teachers and students, potentially limiting the flexibility in teaching practices and provided individualised support to students with WM difficulties.

> P15: *"You have to do so much in education that sometimes you think. . .I'm overlooking all these things. But if I do that, it will eventually get stuck. You think, yes, it's really important to keep repeating that properly. While deep down, you know that you have to move on to the next thing you have to do."*

Another influencing factor is the support teachers receive from professionals and other colleagues. One teacher expressed feeling insecure when working with students with learning difficulties, but highlighted valuing the support she receives from others.

P3: *"But with others, I keep searching for another way, and then I do like to ask my colleagues for help, like how do you do it. Or to approach the CLB (Centre for Student Guidance) and ask them to look at it from a different perspective. What also makes it good is the support network that we can rely on if parents approve. Because sometimes, you just don't know anymore."*

### 3.2.4. Other Factors

**Parents/home situation.** Parents often have different expectations for their children and may not use the same approaches at home compared to the teachers. This may be a barrier to the effective implementation of teaching strategies aimed at supporting children's WM, mainly due to the lack of consistency. In many instances, parents tend to adopt a more lenient approach towards their children and provide additional support, whereas teachers prioritise improving independent thinking skills.

P10: *"Because if it's always 'The teacher who says, the teacher will tell me anyway, and at home, mom or dad will also tell me what I should do. So I don't really have to think for myself'. And that's also the case with helping in the classroom—'The teacher will come to help me'. Sometimes I say, 'No, you will do this alone now'."*

Nevertheless, teachers recognise the parents' valuable role in supporting the independence and cognitive development of the child at home.

P5: *"Parents should also be involved. I can see that your child is struggling, and I often don't receive the homework from them. Why do you think that is? How is the situation at home in the evenings? That is also very important. You encounter all these kinds of situations. And then you have to respectfully tell people, 'Oh, I understand that it's indeed difficult. Maybe we should take a look together at how we can help him? Would you be open to checking his agenda with him in the evenings and making his school bag together?'."*

However, there are situations where parents may face difficulties in providing support and collaborating effectively with teachers. This can be due to various home environment factors that contribute to parental stress. A problematic home situation can hinder providing (and receiving) adequate support since the child's WM capacity may already be compromised. Some teachers also highlighted the impact of the family's socio-economic status as a contributing factor to the challenges parents and, in turn, teachers face in supporting their children.

## 4. Discussion

This qualitative study sought to provide a comprehensive overview of the specific TSI strategies implemented by the teachers to support students' WM in the classroom, the underlying beliefs the teachers hold about the effectiveness of these strategies, and the factors influencing the utilisation and efficacy of these strategies.

### 4.1. Specific Strategies

The first goal of the study was to identify which strategies teachers employ in practice in comparison to what is outlined in theory; or, more specifically, which TSI strategies are seen to be particularly valuable and effective when addressing children's WM-related difficulties in the classroom. When fostering a supportive learning environment that can enhance children's WM, teachers employed strategies from all three domains (i.e., instructional support, classroom organisation, and emotional support, as outlined in the TTI framework [75]), underscoring the applicability and effectiveness of each type of support when assisting students with WM difficulties. Furthermore, the results of the interviews provided strong support for the relevance of numerous dimensions and various indicators for WM improvement. Wherever possible, specific strategies were matched with the aspects mentioned in the CLASS, highlighting those most relevant for children with WM difficulties exhibited in the classroom (for an overview, see Figure 4).

| DOMAINS | DIMENSIONS | CLASS INDICATORS | IDENTIFIED STRATEGIES by the teachers to support WM |
|---|---|---|---|
| INSTRUCTIONAL SUPPORT | CONCEPT DEVELOPMENT | analysis and reasoning | critical thinking |
| | | integration | |
| | | creating | |
| | | connections to the real world | |
| | QUALITY OF FEEDBACK | scaffolding | providing instruction |
| | | feedback loops | providing feedback |
| | | prompting thought processes | prompts and cues |
| | | providing information | |
| | | encouragement and affirmation | |
| | LANGUAGE MODELLING | frequent conversation | |
| | | open-ended questions | |
| | | repetition and extension | |
| | | self- and parallel talk | self- and parallel talk |
| | | advanced language | |
| | | | *rules* |
| | | | *modelling* |
| | | | *increasing challenge* |
| | | | *adjusting support* |
| CLASSROOM ORGANISATION | BEHAVIOUR MANAGEMENT | clear behaviour expectation | behaviour regulation |
| | | redirection of misbehaviour | |
| | | proactive | monitoring |
| | | student behaviour | |
| | PRODUCTIVITY | maximising learning time | |
| | | routines | routines |
| | | transitions | providing an overview |
| | | preparation | |
| | INSTRUCTIONAL LEARNING FORMATS | effective facilitation | |
| | | variety of modalities and materials | varying of modalities and materials |
| | | student interest | promotion of student interest |
| | | clarity of learning objectives | |
| | | | *classroom arrangement* |
| EMOTIONAL SUPPORT | POSITIVE CLIMATE | relationship | emotional safety |
| | | positive affect | |
| | | respect | |
| | | positive communication | attuned communication |
| | NEGATIVE CLIMATE | negative affect | |
| | | punitive control | |
| | | sarcasm/disrespect | |
| | | severe negativity | |
| | SENSITIVITY | responsiveness | responsiveness |
| | | awareness | sensitivity |
| | | addresses problems | |
| | | student comfort | |
| | REGARD FOR STUDENT PERSPECTIVE | flexibility and student focus | |
| | | support for autonomy and leadership | autonomy, leadership, and responsibility |
| | | student expression | student expression |
| | | restriction of movement | |

**Figure 4.** The TTI Framework [75], CLASS Indicators [81], and Teacher-Identified Strategies for Supporting Students' WM.



Based on the information elicited from the interviews, teachers' interactions when supporting children exhibiting poor WM or associated difficulties can be understood through the triadic framework of TTI [75]. However, when applied to students with WM problems, the findings of the current study make a unique contribution and suggest that these three types of support can, alternatively, be arranged in a hierarchical order, in terms of their relevance and specificity for tackling WM-related challenges [103]. As evident in the teacher interviews, out of the three types of support, strategies falling under emotional support seem to be the least WM-specific—not directly supporting children's WM or addressing WM-related problems. Instead, the identified strategies allow teachers to create a positive climate in the classroom and to promote (or improve) a trusting, open, and warm relationship between the teacher and the child. In turn, and through various means, this ensures a secure and safe environment, conditions necessary for sustainable and lasting change in improvements in children's WM-related behaviour [104]. More importantly, the previous literature suggests that such an environment is important for a high-quality dyadic relationship (i.e., low levels of conflict and high levels of closeness) between the teacher and the child [105,106]. These studies highlight that children's learning and cognitive development (including WM) might not benefit from the TSI's quality and strategies when the dyadic relationship is poor. The results obtained from the thorough analysis of the teacher interview data indicate that, following emotional support, classroom organisation strategies come into play, emphasising the importance of a structured and orderly environment in the classroom, removing unnecessary distractions and 'noise' interfering with the implementation of instructional strategies [107]. Classroom organisation strategies can be considered as an approach falling between direct and indirect support, ensuring optimal conditions for cognitive functions, including WM, to operate effectively [6,71]. The (in)direct influence of a well-organised classroom environment on the efficacy of the instructional strategies is, therefore, highlighted. Finally, at the top of the hierarchy, with almost half of all employed strategies as identified by the teachers, lie those of instructional support. Such strategies directly support, challenge and train/improve children's WM abilities. These strategies are not just about aiding students in the immediate context, but also about equipping them with skills for future challenges. Such findings suggest that this type of support might be particularly appropriate and effective when supporting children with poor WM and dealing with children's WM-related difficulties. Overall, the holistic approach, taking into account both direct and indirect approaches, offers a comprehensive understanding of the multifaceted challenges faced by students with WM difficulties and the diverse strategies teachers can employ to address them.

Regarding specific indicators, most of the identified strategies by the teachers could be matched with those outlined by the CLASS [81]. In addition, some new (i.e., not specified in the CLASS) indicators were distinguished. As introduced in Figure 4, these findings are presented by grouping specific indicators based on the type of support provided: (1) instructional support, (2) classroom organisation, and (3) emotional support. However, teacher interviews allowed for identifying most WM-relevant indicators and those seemingly not applicable when supporting WM, as perceived by the teachers. Therefore, the strategies within each domain are ordered according to their aim, directly or indirectly targeting and supporting WM.

Firstly, concerning instructional support, teachers highlighted certain strategies as particularly beneficial for children with WM-related difficulties. These strategies (in line with the CLASS) included critical thinking, providing instruction, providing feedback, and prompts and cues. The importance of these strategies is also indicated in the literature on recommended practices in primary education [108–111]. Based on the literature (and the CLASS), repetition and extension can also be deemed relevant, given that repetition aids in reinforcement in memory, and extension helps bridge between known information and new knowledge, making it easier for WM to integrate and process [112]. Although not outlined as a separate strategy in the teacher interviews, repetition was somewhat integrated by the teachers into prompting (i.e., repeating information in unison with the

children) and into self- and parallel talk. It was considered one of the most beneficial strategies teachers can employ with students with WM problems, leading to better information retention. Another effective strategy outlined by the teachers (but not mentioned in the CLASS) for enhancing students' WM is increasing challenge. This strategy actively engages children's WM, challenges it, and, in turn, strengthens it. Some components of increasing challenge can be compared to those of scaffolding (i.e., fading out the support or instructions provided), which, in combination with the gradual building up of the difficulty level of tasks/assignments, continuously activate and reinforce students' WM [113]. In addition, a strategy of adjusting support to a student's unique needs was seen as crucial for children with WM difficulties. Adjusted support can target weak areas, gradually improving WM capacities over time, allowing for modifications and alternative ways to approach tasks, bypassing WM limitations and avoiding overwhelming the student with demands [114,115]. Even though some overlap with scaffolding could be assumed (i.e., facilitating independence by providing additional assistance and tools), tailored support offers various approaches and is inherently more personalised, addressing the individual needs of a student rather than using a general supportive framework. The idea of adjusted support, not only as a unique strategy, but as an approach applied when employing each of the strategies, was highlighted as crucial.

Other strategies emerged from the teacher interviews but were considered as indirectly supporting children's WM. One of the indicators, defined rules, although included in the CLASS as methods for preventing and redirecting misbehaviour, differed in definition in the teacher interviews. Rules were highlighted as techniques referring to the instruction following and task completion. These can be particularly beneficial for children with poor WM, as rules provide a clear structure and reduce the cognitive load by offering predictable and easily followed guidelines and expectations [116]. Similarly, modelling reduces the cognitive demands on WM by offering a visual guide to follow and, over time, building skills. Children with WM challenges can focus on one step at a time without having to hold the entire process in memory. Some other strategies did not seem to emerge during the teacher interviews. Although the literature (in agreement with CLASS) suggests that such strategies are relevant, at least in a classroom context, when focusing on a typical student, teachers not mentioning certain strategies point to the idea that they are not considered as directly related to WM. These strategies include advanced language, creating, and connections to the real world. They can be considered as not addressing immediate WM processing challenges directly nor alleviating the demands on WM. Furthermore, they might even introduce an additional cognitive load. For instance, advanced language use can challenge and expand linguistic structures; however, for students with WM difficulties, it might introduce unnecessary complexity and confusion [117,118]. Similarly, creating or creative processes, while beneficial for cognitive flexibility, might be straining WM [119,120]. Finally, connections to the real world (or, in other words, relating concepts to students' actual lives) could be perceived as adding complexity to lessons and, in turn, overwhelming the students with WM difficulties instead of aiding them. While using real-world applications can be an effective teaching strategy for most students, it may require careful adaptation to suit the needs of students with WM problems. These strategies, therefore, might be better applied with children who do not show WM-related difficulties in the classroom or applied later on, when children's initial WM challenges have been addressed and tackled. In conclusion, the strategies that emerged from teacher interviews provide unique insights on components and strategies especially beneficial for supporting children's WM and dealing with children's WM-related difficulties in the classroom.

Secondly, regarding classroom organisation, teachers identified various strategies as valuable and effective for improving children's poor WM. Identified strategies, including behaviour regulation, routines, providing an overview, and the varying of modalities and materials, could be directly matched with those outlined in the TTI framework and the CLASS as falling under the three classroom organisation dimensions. The literature suggests the potential effectiveness of these strategies [71,121–123]. The clarity of learning

objectives (outlined in the CLASS, but not explicitly mentioned in the interviews) could also be helpful, given that children with WM difficulties benefit from understanding the primary goals of a lesson, which helps them compartmentalise information and focus their cognitive efforts. However, this concept was already somewhat covered by rule application in the classroom and, thus, did not emerge as a separate unique strategy. Although suggested by the CLASS as effective for student learning, other indicators were either not mentioned during the interviews or merged with other strategies in teacher answers and, therefore, these were grouped into broader encompassing strategies. This (lack of) finding suggests that these strategies might be less WM-relevant/focused (but still important for learners). Such strategies include student behaviour and effective facilitation, although the indicator referring to student interest was identified. These strategies can be seen as not addressing immediate WM processing challenges directly nor alleviating the demands on WM (i.e., connections to the real world, self- and parallel talk). Similar to instructional support strategies, these do not directly address WM challenges or support WM. In particular, monitoring (highlighted by the teachers, but not mentioned in the CLASS) could be seen as one of the strategies not directly addressing WM challenges. Instead, it is a proactive measure applied as a preventive and detective strategy, used for identifying children's individual challenges and needs, but not directly impacting children's WM [124,125]. Other strategies, instead, seem to play a role in creating a supportive learning environment and contribute to an overall structured, engaging, and positive classroom experience, which can indirectly support all learners, including those with WM difficulties. This, indeed, is especially highlighted by one new strategy, not covered by the CLASS, that emerged from the teacher interviews. This strategy referred to the overall classroom arrangement, which can be simply defined as the structuring of the physical environment (i.e., desk arrangement) and the organisation of materials. The latter, however, has some overlap with preparation as defined in the CLASS (i.e., having materials ready and accessible to the students). The teachers emphasised that having an organised learning environment, in turn, relieves students' WM and effectively supports the children by providing easy access to the necessary materials and reducing unnecessary distractions [126,127].

Thirdly, considering emotional support, teachers stressed some strategies as relevant when supporting children with WM difficulties. These strategies (also part of the CLASS) included responsiveness, sensitivity, student expression, and autonomy, leadership, and responsibility. The importance of these strategies is highlighted in the literature [10,128–130]. Flexibility and student focus, although not explicitly mentioned by the teachers (but covered by the CLASS), can be traced back to some of the other identified strategies (e.g., adjusting support and the varying of modalities and materials). Other indicators, such as emotional safety, attuned communication, and student comfort, though still important for general well-being (as outlined in the CLASS), were somewhat less WM-relevant (and, therefore, not mentioned in the teacher interviews). Instead, all these indicators contributed to an environment that is warm, encouraging, and uplifting, which can help to reduce anxiety and stress, known to impair WM [131–133].

In conclusion, various strategies have been proposed by the literature as improving the learning environment and learners' outcomes. The findings from the teacher interviews facilitated the streamlining of these strategies and highlighted the (direct and indirect) strategies most effective and beneficial when assisting children with poor WM. In addition, a direct comparison with the CLASS [81] enabled the identification of specific indicators most applicable for students with WM-related difficulties (in comparison to all learners) in primary school classrooms. Most of the identified strategies by the teachers could be matched with those outlined; nevertheless, some new indicators were mentioned, highlighting the discrepancy between theory and practice.

*4.2. Underlying Theories*

The second goal of this study was to explore the theories or rationales for identified strategy use in the classroom. Or, more specifically, the underlying beliefs the teachers

hold about the effectiveness of these strategies, in comparison to what is outlined in the literature. Generally, teachers seemed to be aware of some of the underlying mechanisms for the strategies used. However, when prompted to elaborate, significant differences highlighting the gap between theory and practice were seen.

Regarding teachers' rationale for the efficacy of each of these strategies, support for the sociocultural theory was evident. The teachers, on numerous occasions, highlighted the importance of the gradual release of support provided, scaffolding instructions given, and fostering discussions (by means of posing open-ended questions) in order to guide and facilitate students to build and internalise knowledge acquired and, in turn, overcome the challenges posed by their WM difficulties. Furthermore, the attachment theory was identified and endorsed by the interviewed teachers. The teachers recognised the value of fostering a secure learning environment, which, in turn, allows students to express their concerns and seek guidance when needed. The findings from the conducted interviews, in addition, revealed some shortcomings of the current teacher knowledge. Self-determination theory was not explicitly identified. Even though some of the principles belonging to this theory were incorporated in the teachers' reasoning (e.g., the promotion of active engagement and ownership in the learning process), the teachers proposed alternative rationales instead. Similar patterns were seen regarding social learning theory. Although the teachers described some aspects grounded in social learning theory (e.g., the importance of attention for successful retention and reproduction of the (expected) behaviour), no explicit reference to this theory as a potential rationale was reported.

Importantly, some ideas (not outlined as commonly in the literature in relation to the TTI framework) emerged in the teacher interviews. The teachers highlighted the principles of cognitive load theory [102] on several occasions. Cognitive load theory is concerned with the limited capacity of students' WM, the inherent complexity of the material (intrinsic load), the manner of presentation (extraneous load), and the cognitive effort required for constructing knowledge (germane load). By aligning strategies with principles from cognitive load theory, teachers present information in a manner that reduces the unnecessary cognitive load, allowing students to focus on processing and understanding the essential content. Furthermore, the teachers mentioned ideas seemingly in line with a skills-building approach. This approach, in education, emphasises the idea that knowledge is not sufficient for success in learning and that skills are necessary for applying and transferring the knowledge into practice [134]. Both of these ideas were indirectly referred to when discussing the strategies falling under all three domains of support.

More importantly, aspects of cognitive load theory, a skills-building approach, in combination with principles of self-determination and social learning theories, emerged in three teacher-identified rationales. (1) Raising and improving attention and focus resonates with principles from the cognitive load theory, which addresses the reduction of the extraneous cognitive load and the improvement of the germane load to support learning. Furthermore, raising attention and focus can lead to the development and improvement of several important skills, such as listening skills—to better understand spoken instructions, follow conversations, and engage in discussions and time management skills—to efficiently allocate time to tasks [135,136]. Additionally, the emphasis on enhancing motivation, evident in the teacher interviews, and raising attention and focus can lead to the development and improvement of important skills, and seem to draw parallels with self-determination theory, which encourages the promotion of active engagement and ownership in the learning process [137]. Furthermore, (2) incorporating repetition and establishing routine in the classroom, as an underlying rationale, align well with the principles of cognitive load theory (and, to a lesser extent, the tenets of social learning theory). When instructions are repeated, children have more opportunities to absorb the content, leading to the repeated activation of their WM and stronger retention. Additionally, repetition helps reduce the cognitive load on children's WM as the instructions become routine (through automatisation). Moreover, repeated exposure to and practice of a specific task allows children to become proficient in that task over time. By mastering one task, students can build upon that

foundation to learn more complex skills. Finally, teachers highlighted the importance of (3) the development of metacognitive skills. Such skills can include strategic planning, goal setting, self-monitoring, decision making, and problem solving, all of which are essential for children to become independent learners [138]. Encouraging children to think independently and reflect on their own learning resonates with self-determination theory, which links autonomy to increased motivation and engagement [139]. Additionally, by promoting self-reflection, teachers support the development of metacognitive skills, aligning with the principles of social learning theory. Some interviewed teachers also highlighted that aiding students in recognising their own forgetfulness can enhance metacognitive skills. Drawing from cognitive load theory, it is evident that by providing strategies to counter forgetfulness, teachers can mitigate cognitive overload, further bolstering learning outcomes, including WM.

In conclusion, teachers' rationale for strategy use often aligns with the proposed theories. The teachers were able to identify and elaborate on the key principles of some theories, while for others, the teachers made implicit references to the ideas in line with the theories. Such findings highlight the teachers' somewhat limited understanding or at least their lack of terminology of the underlying mechanisms behind each of the strategies used. There seems to be a discrepancy between theoretical knowledge and their practical application, highlighting the need for continued professional development and unifying the language/terminology used by the teachers, the trainers, and the researchers. By enhancing the teachers' understanding of educational theories, they can be better equipped to apply strategies effectively. However, the teachers also offered a unique view of the rationales by identifying new explanations. The teachers' rationales and insights underscore the need to integrate them into educational theory, highlighting the invaluable expertise teachers' experiences bring about. Recognising that teachers often draw from diverse theoretical foundations, it is essential to advocate for a collaborative approach to education. This approach should merge elements from various theories and actively seek teacher input, ensuring that classroom experiences inform and enrich the theoretical framework.

### 4.3. Personal Characteristics and Contextual Factors

The third and final goal of the current study was to examine the factors that influence the utilisation and efficacy of these strategies. Teachers, during the interviews, identified various factors falling under the first, second, and third levels of the multilevel supply–use model [88], covering almost all the concepts outlined. Guided by this model, the current study provided a valuable examination of multiple influencing factors shaping student learning, ranging from individual characteristics to broader social and institutional contexts, and highlighted the multi-faceted and interactive nature of educational processes and outcomes, as identified by the teachers. In the context of TSI strategies and WM performance, individual differences in cognitive abilities, temperament, and socio-emotional skills emerged as particularly important characteristics affecting how students respond to TSI strategies [140,141]. For example, a child with a high WM capacity may be better able to engage in complex learning activities, while a child with a lower WM capacity may benefit more from structured and scaffolded interactions [45]. Furthermore, the teachers stressed that their own beliefs, attitudes, and professional competence can significantly influence their implementation of TSI strategies [142]. For example, teachers who understand (or are sufficiently informed and trained in) the role of WM in learning may be more likely to incorporate strategies that support WM in their instruction. Although seen as less influential, teachers mentioned that peer interactions could further impact a child's learning experience [143,144]. Peers can serve as models, provide feedback, and offer emotional support (similar to teachers), all of which can impact a child's engagement and learning outcomes [145,146]. In addition, the organisation and management of the classroom, as well as the emotional climate, can affect students' engagement, motivation, and, ultimately, their learning outcomes [147]. A well-structured and supportive classroom environment can reduce the cognitive load and support WM performance [148]. Furthermore, less WM-

specific, but, nevertheless, relevant broader school factors are important for the success and efficacy of the strategy implementation by the teacher. As evident in the teacher interviews, school culture, provided resources, and policies can influence the implementation and effectiveness of TSI strategies [149,150]. For example, a school culture that values and supports the teachers' professional development can enhance the effective implementation of TSI strategies [151]. Outside of the classroom and school environment, parents and families can further impact children's learning through their expectations, involvement, and support at home [152,153]. For example, parental involvement in learning activities can reinforce TSI strategies used in the classroom and support WM performance further [86]. Parents and family might be of particular importance when considering maintaining children's WM improvements, as durable effects are most likely when support is provided consistently and across diverse contexts.

In conclusion, the identified factors highlight the interconnected and multilevel nature of the factors affecting learning (in line with Brühwiler and Blatchford's model [88]). The teacher interviews highlighted that the efficacy of TSI strategies is not solely reliant on the strategy itself, but is significantly influenced by individual capacities, teacher characteristics, and, in an indirect way, by broader contextual factors. These findings suggest that to effectively support and improve children's WM performance, it is important to consider numerous factors concurrently.

*4.4. Limitations*

While this qualitative research, by delving deep into teachers' perspectives and experiences, helped to enrich the understanding of teacher-applied strategies for supporting students' WM and offered insights into the settings and contexts in which these interactions occur, there are some limitations to consider.

One potential limitation is the selection bias stemming from participant recruitment. It is possible that the teachers, who were willing to collaborate and participate in the study, already had an increased interest or focus on WM (challenges) in their classroom. Therefore, the conclusions regarding the strategies that are already being implemented in primary school classrooms, as well as teachers' awareness and knowledge regarding the theories and rationales behind their teaching strategy use, should be interpreted with caution and might not be representative of the broader teaching community. Nonetheless, the current findings reveal the potential, and such insights remain pertinent even if there would be a bias.

To build upon the current work, a mixed-method approach, incorporating both qualitative and quantitative data, could offer a more comprehensive perspective. Further empirical investigation, specifically examining the efficacy of the identified strategies on children's WM through standardised (pre- and post-) assessments, would strengthen the current findings. Such a study could pave the way for well-informed interventions tailored to strengthening and improving children's WM through teaching practices.

**5. Conclusions**

This qualitative study, which explores TSI strategies for addressing students' WM-related difficulties in the classroom, has notable implications for both theory and practice. By employing a dual perspective, combining deductive, theory-based insights from the TTI framework and an inductive practice-based perspective derived from teachers, this study offers a detailed understanding of strategies applied in real-world classroom settings. It provides educators with a practical guide detailing TSI strategies grounded in theoretical frameworks and empirical experiences. These findings underline the most beneficial strategies for students with WM challenges and recommend their integration into teacher training programmes for a more focused and tailored approach. Moreover, the study emphasises the significance of a student-focused approach when implementing TSI strategies. The success of these strategies is highly dependent on individual student attributes, needs, and abilities, alongside external factors, such as teacher-related aspects

and environmental conditions. The current findings highlight that teachers and schools should actively engage with parents, educating them about the importance of WM and how the parents can support their children at home, resulting in a collaborative approach.

In conclusion, the current study provides a comprehensive overview of TSI strategies employed by primary school teachers to enhance students' WM and manage WM-related behaviour in the classroom. To address the multifaceted challenges faced by students with WM difficulties, a holistic approach is recommended. Teacher training programmes should equip teachers with evidence-based theoretical knowledge and practice-based strategies, and, in turn, teachers should be given the opportunity to provide feedback on the strategies they are trying out, what is working for which children, and what is not, while a collaborative approach between all parties can inform classroom practices, professional development, and policy-making.

**Author Contributions:** Conceptualisation, M.H. and D.B.; methodology, M.H., N.D.V., E.H. and D.B.; data curation, N.D.V. and E.H.; formal analysis, S.S., S.P. and E.H.; writing—original draft preparation, S.S.; writing—review and editing, S.S., S.P., C.X., M.H. and D.B.; visualisation, S.S.; supervision, M.H. and D.B.; project administration, D.B. All authors have read and agreed to the published version of the manuscript.

**Funding:** This research study was funded by the KU Leuven Internal Funding (C14/19/052).

**Institutional Review Board Statement:** The study protocol was approved by the Social and Societal Ethics Committee of KU Leuven (G-2019-1320).

**Informed Consent Statement:** Informed consent was obtained from all subjects involved in the study.

**Data Availability Statement:** The data presented in this study are available on request from the corresponding author. The data are not publicly available due to privacy restrictions.

**Conflicts of Interest:** The authors declare no conflict of interest.

## Appendix A. The Classroom Assessment Scoring System (Adapted from [81])

| INSTRUCTIONAL SUPPORT | | | |
|---|---|---|---|
| | CONCEPT DEVELOPMENT | Analysis and reasoning: | why and/or how questions; problem solving; prediction/experimentation; classification/comparison; evaluation |
| | | Creating: | brainstorming; planning; producing |
| | | Integration: | connects concepts; integrates with previous knowledge |
| | | Connections to the real world: | real-world applications; related to students' lives |
| | QUALITY OF FEEDBACK | Scaffolding: | hints; assistance |
| | | Feedback loops: | back-and-forth exchanges; persistence by teacher; follow-up questions |
| | | Encouragement and affirmation: | recognition; reinforcement; student perspective |
| | | Providing information: | expansion; clarification; specific feedback |
| | | Prompting thought processes: | asks students to explain thinking; queries responses and actions |
| | LANGUAGE MODELLING | Frequent conversation: | back-and-forth exchanges; contingent responding; peer conversations |
| | | Open-ended questions: | questions require more than a one-word response; students respond |
| | | Repetition and extension: | repeats; extends/elaborates |
| | | Advanced language: | variety of words; connected to familiar words and/or ideas |
| | | Self- and parallel talk: | maps own actions with language; maps student action with language |

| | | | |
|---|---|---|---|
| **CLASSROOM ORGANISATION** | **BEHAVIOUR MANAGEMENT** | Clear behaviour expectation: | clear expectations; consistency; clarity of rules |
| | | Proactive: | anticipates problem behaviour and escalation; low reactivity; monitors |
| | | Redirection of misbehaviour: | effective reduction of misbehaviour; attention to the positive; uses subtle cues to redirect; efficient redirection |
| | | Student behaviour: | frequent compliance; little aggression and defiance |
| | **PRODUCTIVITY** | Maximising learning time: | provision of activities; choice when finished; few disruptions; effective completion of managerial tasks; pacing |
| | | Routines: | students know what to do; clear instructions; little wandering |
| | | Transitions: | brief; explicit follow-through; learning opportunities within |
| | | Preparation: | materials ready and accessible; knows lessons |
| | **INSTRUCTIONAL LEARNING FORMATS** | Variety of modalities and materials: | range of auditory, visual, and movement opportunities; interesting and creative materials; hands-on opportunities |
| | | Student interest: | active participation; listening; focused attention |
| | | Clarity of learning objectives: | advanced organisers; summaries; reorientation statements |
| | | Effective facilitation: | teacher involvement; effective questioning; expanding children's involvement |
| **EMOTIONAL SUPPORT** | **POSITIVE CLIMATE** | Relationship: | physical proximity; shared activities; peer assistance; matched affect; social conversation |
| | | Positive affect: | smiling; laughter; enthusiasm |
| | | Respect: | eye contact; warm, calm voice; respectful language; cooperation and/or sharing |
| | | Positive communication: | verbal affection; physical affection; positive expectations |
| | **NEGATIVE CLIMATE** | Negative affect: | irritability; anger; harsh voice; peer aggression; disconnect or escalating negativity |
| | | Punitive control: | yelling; threats; physical control; harsh punishment |
| | | Sarcasm/disrespect: | sarcastic voice/statement; teasing; humiliation |
| | | Severe negativity: | victimisation; bullying, physical punishment |
| | **TEACHER SENSITIVITY** | Awareness: | anticipates problems and plans appropriately; notices lack of understanding and/or difficulties |
| | | Responsiveness: | acknowledges emotions; provides comfort and assistance; provides individualised support |
| | | Addresses problems: | helps in an effective and timely manner; helps resolve problems |
| | | Student comfort: | seeks support and guidance; freely participates; takes risks |
| | **REGARD FOR STUDENT PERSPECTIVES** | Flexibility and student focus: | shows flexibility; incorporates students' ideas; follows students' lead |
| | | Support for autonomy and leadership: | allows choice; allows students to lead lessons; gives students responsibility |
| | | Student expression: | encourages student talk; elicits ideas and/or perspectives |
| | | Restriction of movement: | allows movement; is not rigid |

**Appendix B. The Coding Template**

| RQ1: Specific strategies | | |
| --- | --- | --- |
| Theme | Subthemes | Codes |
| INSTRUCTIONAL SUPPORT | | |
| Concept development | | |
| | Critical thinking | - (self-) reflection<br>- analysing and reasoning<br>- comparing<br>- solution-oriented thinking |
| Quality of feedback | | |
| | Providing instruction | - breaking down into smaller steps or pieces<br>- concrete, simple, short instructions<br>- limited number of tasks, assignments, or instructions at a time<br>- repeating<br>- structuring |
| | Providing feedback | |
| | Prompts and cues | - giving a hint or tip |
| Language modelling | | |
| | Self- and parallel talk | |
| | *Rules* | |
| | *Modelling* | - modelling by teacher |
| | *Increasing challenge* | |
| | *Adjusting support (individualised approach)* | - remediation<br>- compensation<br>- differentiation<br>- stimulation<br>- dispensation |

| Theme | Subthemes | Codes |
| --- | --- | --- |
| CLASSROOM ORGANISATION | | |
| Behaviour management | | |
| | Behaviour regulation | - clear behavioural expectations and assignments<br>- redirection of misbehaviour |
| | Monitoring | |
| Productivity | | |
| | Routines | |
| | Providing an overview | |
| Instructional learning formats | | |
| | Varying of modalities and materials | |
| | Promotion of student interest | - raising (intrinsic) motivation<br>- raising attention, focus, and active listening<br>- raising concentration (avoiding distractions) |
| | *Classroom arrangement* | - order and material organisation |

| Theme | Subthemes | Codes |
|---|---|---|
| EMOTIONAL SUPPORT | | |
| Positive climate | | |
| | Emotional safety | - creating openness to talk about emotions and problems<br>- making mistakes is okay<br>- not belittling a child in front of class<br>- staying calm and patient, not getting angry |
| | Attuned communication | - complimenting or praising<br>- limiting negative communication on classroom level<br>- using humour |
| Sensitivity | | |
| | Responsiveness | - communicating about emotional availability<br>- discussing problems, needs, or concerns with the child<br>- encouragement and affirmation<br>- experiencing success<br>- giving the child confidence<br>- showing understanding |
| | Sensitivity | - awareness of individual needs and differences<br>- noticing distress<br>- noticing students seeking for support and guidance |
| Regard for student perspective | | |
| | Autonomy, leadership, and responsibility | |
| | Student expression | |

| RQ2: Rationales | | |
|---|---|---|
| Theme | Subthemes | Codes |
| Sociocultural theory | | |
| | Scaffolding | |
| | Challenging learning activities | |
| | Matching the learning style of the child | |
| Attachment theory | | |
| | Safe haven | |
| | Secure base | |
| Metacognition | | |
| | Self-knowledge | |
| Attention and focus, & motivation | | |
| | Intrinsic motivation | |
| | Extrinsic motivation | - learning through disappointment<br>- learning through frustration |
| Repetition and routine | | |
| | Automatisation | |
| | Repeated activation | |
| | Reducing cognitive load | |

| RQ3: Influencing factors | | |
| --- | --- | --- |
| Theme | Subthemes | Codes |
| Child characteristics | | |
| | Child characteristics | |
| | Individual learning preconditions | |
| | Individual learning processes | |
| Teacher characteristics | | |
| | Teaching competence | |
| | General characteristics | |
| Learning environment | | |
| | Peers | |
| | Classroom context | |
| | School context | |
| Other factors | | |
| | Parents | |
| | Home situation | |

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
