# Peer review of "A Qualitative Study into Teacher–Student Interaction Strategies Employed to Support Primary School Children’s Working Memory"

_education, doi:10.3390/educsci13111149_

Round 1

Reviewer 1 Report

Comments and Suggestions for Authors

I would like to thank for the opportunity to review this manuscript. It reports on a very interesting and relevant study that is strongly embedded in theory and uses a very systematic qualitative approach to unravel what kind of strategies teachers use to support students (weaker) working memory abilities. The qualitative approach adds to the current literature, and provides insights that can feed future research and practice. The paper has great potential. In my opinion, several issues require attention to make this paper suitable for publication.

Although the paper is generally well-written, the main issues that need improvement relate to (1) writing more concise to make the paper more manageable for the reader, and (2) stronger line of reasoning (especially in the second part of the introduction), and (3) improved structure in the results section. In addition, (3) some findings either need clarification or perhaps relabeling.

Introduction

The authors have written a very well-developed introduction section, that includes the discussion of several relevant theoretical frameworks. This sound theoretical grounding is a strong asset of this study. However, the introduction is also a bit long, and the reasoning in the final sections could be a bit more clear and to-the-point. Below, I give some more specific suggestions for improvements:

1. Op page 3, in the section on ‘Working Memory and Education’, the authors describe how WM affects learning and academic achievement. However, more recently, the bidirectional relation between learning and working memory has received more attention. There are some studies that show that learning activities and content knowledge also affect WM development (see for example Kahl et al, 2022; Peng & Kievit, 2020). This is also relevant for the argumentation that previous WM training programs were often delivered out of context (line 146 on page 3).

2. On page 6, the paragraph on the CLASS instrument as an operationalization of the TTRO framework seems to hamper the line of reasoning a bit. I is not clear if and how this instrument is crucial to mention here and seems a bit out of place. If it is mentioned because it is used as a source to feed the coding scheme used, it would probably fit better in the method section. Perhaps it is described here to introduce the observational studies on page 7, paragraph 1? In that case, the description may be a bit more concise and more explicitly connected to that paragraph. Figure 2 could perhaps be moved to an appendix.

3. The order and line of reasoning in the last section of the introduction (page 7 and 8) could be developed further to guide the reader in understanding better what this study adds.

3.1. On page 7, the first paragraph mentions different characteristics of children and teachers that could be relevant of the relation between TSI and WM. This paragraph seems to be used as substantiation or introduction to the multilevel supply-use model in the next paragraph. However, the core message and line of reasoning is not yet clear. The paragraph ends with: “ Given the limited findings, individual and contextual characteristics, impacting the use and efficacy of the strategies and influencing the achieved outcomes, should be further explored.” The sentence is quite long making it hard to understand. It would also help if the authors clarify what about these characteristics needs further exploration. Is the aim to identify which characteristics play a role? How they interact?

3.2. In the description of the multilevel supply use model, the connection to the specific topic of the current study (TSI and WM) is not made very explicit. Explaining this more, perhaps by using examples, may help to clarify what this model adds and how it will be used in the study. It is mentioned for example (line 284-285 on page 7) that  “ the multilevel aspects of the model acknowledges that both TSI and student learning outcomes can vary within classrooms as well as between classrooms…” It is not explained why this is relevant or which gap it fills. Which levels will this study focus on? This relevance of the multilevel supply use model is mentioned on Page 8 (line 310-314) however. Perhaps this could be moved to an earlier part of this section.

3.3. The gap in research seems to be discussed in the next section on page 8, although the gap is a bit implicit. A more explicit statement on what the gap in literature is, will make the relevance of this study more clear. I think that reshuffling the order here a bit (starting with the gap, then presenting the model as a solution) could help the reader in understanding how this particular model could help to fill the gap in knowledge. That would also make it more clear why interviews fit well with the research aims.

3.4. On page 8, line 301-303 it says: “ It is, therefore, important to better understand the strategies teachers apply in their classrooms to improve or effectively add to their teaching practices through interventions.” It is not clear how that connects to the previous sentence, where it is argued that interviews give insight in teacher experiences.

Method section

The author use a very systematic approach to the coding of interviews, and give a very detailed and stepped description, which will make it possible to replicate this study.

4. The section on Template analysis (page 11-13) is quite wordy. I see several opportunities to write more concisely, without losing information. If the authors put in that effort, it will be easier to follow by the reader. Below are some additional comments:

4.1. The description of the multilevel supply use model (page 11, line 431) in the Template analysis section, seems to suggest that only the ‘supply’ element is used in coding, and not the ‘use’ element. Can the authors clarify this?

4.2. On page 12 (line 454), interrater reliability is discussed and it is explained quite elaborately which criteria are used, but I think the actual results are not mentioned. What was the interrater reliability for the 4 interviews that were double coded?

4.3. On Page 12, under Step 4 the description of the use of the TTI framework could be more to-the-point. The statement (line 474-475): “ Within each support dimension, two to four different levels of coding were used”  is a bit ambiguous. I think it refers to the dimensions that were identified in the introduction section, but that is not clear yet. The same goes for levels of coding in groups (line 478-479)

Results

The authors made an effort in describing the findings in a structured way. Nevertheless, I sometimes found it a bit difficult to follow how some results are connected to other parts of the manuscript, but also to other parts of the results. Below, I identify some issues and/or give suggestions for improvement:

5. The connection with other parts of the manuscript could be made more by using consistent language throughout the paper. For example, in the introduction the authors use words as “ the first goal”, while in the results section it is stated ‘ the primary objective”  and the research questions are also worded slightly different, for example: “The first goal is to identify specific TSI strategies employed by the teacher to support their students WM and mange WM related problematic behavior sin the classroom” (page 9) versus “The primary objective of this study was to obtain a comprehensive overview of the  specific strategies implemented by teachers in primary school classrooms, particularly focusing on children with WM problems and related difficulties” (page 13). Although repetitive wording can get boring, in certain elements it repetition can have a deliberate function of structuring.

6. The structure of Table 2 does not seem to align with the order in which certain findings are discussed in the text. The rationales on pages 25-27 for example, seem to be in the wrong place: the mechanisms ‘Metacognition’,  ‘Attention and focus & (intrinsic) motivation’  and ‘ Repetition  and routine’ are not discussed in the text under those specific theories and strategies where teachers identified them? While earlier findings were structured along the ‘rationale’  and then ‘ strategies’  subheadings, these sections do have a ‘rationale’  heading but not a ‘ strategies’ heading.

7. Although TTI was used as a framework in the coding, it was not mentioned as part of the results structure or Table 2, which I expected. Was there a reason for this decision? I think it requires some clarification or perhaps an adjustment.

8. The subheading structure in the results section didn’t seem consistent or to follow APA guidelines, which made it harder to see which subsection handled what content. The subheading structure is also not in line with Table 2, where strategies are presented before rationales. Maybe it works better to use the same order in the text: first discussing findings regarding strategies, and then the rationales of the teachers. Another option is to rearrange the table a bit so that it is aligned with how the text is structured. Subheading could be adjusted a bit to give more meaning. The ‘Strategies’  subheading on page 14 could be reformulated to ‘ Scaffolding strategies’  to make it more specific, especially since that subheading is used several times. 

9. Also, the results section included a lot of general descriptions of theories and definitions, which made it quite long. I would keep the definitions and general descriptions of theoretical frameworks and constructs to the introduction. The results section should only focus on the findings.

10. In some parts of the results section, it was difficult to determine what was a finding an what was a more general statement. An example is the statement on page 14 (line 559): “ teachers can provide instructions by breaking them down….”. Based on the formulation it is not clear if this a general statement or a finding. If it is a finding a suggested formulation could be: “Teachers reported providing instructions by breaking down.” 

11. Other parts in de results seem to be more of a conclusion, for example on page 22 (line 935-939): “By providing emotional support in the classrooms, teachers are able to create and maintain a secure and positive environment. […..], and seek guidance when needed.” I would advise not to draw conclusions in the results section.

12. I also have some comments, questions and suggestions regarding more specific findings. Perhaps the authors can either clarify them for the reader or make some adjustments in where these findings fit best or how they are labeled:

12.1. The theme ‘ Increasing challenge’ on page 15 (line 581) seems to fit with ‘ Scaffolding’, there is a strong overlap. I am wondering if these aren’t interchangeable?

12.2. In the ‘ Adjusting support’  theme on page 15 , the authors refer to the ‘zorgcontinuum’  that is used in Belgium, while the study was conducted with Dutch teachers as well. Perhaps a more general description of a tiered approach fits better, and could be more concise. It is perhaps something to elaborate on in the discussion, rather than in the results section.

12.3. On page 19, line 770, the author refer to the use of noise cancelling headphones under the theme ‘ Promotion of student interest’. I am wondering if that could be seen as an example of ‘ Classroom arrangements.’

12.4. On page 19, Providing feedback is discussed as a strategy that fits with self-determination theory, but in the description of findings it does not become clear how it is related to this theory. Does feedback promote feelings of competence and/or autonomy and/or relatedness?

12.5. I wonder if the theme ‘ Student expression’ on page 19 is really a separate theme or if it is part of ‘ Autonomy, leadership and responsibility’

12.6. Based on the description, the theme ‘ Providing an overview’  seems to fit better with scaffolding strategies of cues. It is not clear yet how it relates to self-determination.

12.7. On page 21 (line 873) findings are described regarding the value of facing challenges and frustration. It is not clear to me yet how that relates to social-learning theory.

12.8. ‘Rules’ (page 21) might be discussed (closer) together with ‘ Routines’?

12.9. In the ‘Metacognition’  section, the authors refer to cognitive load theory when discussing findings, but I do not think I saw this theory or construct defined earlier in the manuscript? This might be solved by describing the findings here without referring to constructs such as germane load and extraneous load, and give a more elaborate theoretical discussion in the discussion section.

12.10. The section on ‘ Intrinsic motivation’  on Page 26 seems to fit better with self-determination theory?

12.11. In some sections, it is clearly stated how many teachers provide utterances that fit with a certain theme, while in others (e.g., in ‘Repetition and routine’) this is not indicated, making it harder to weigh the findings.

12.12. On page 27, line 1196, a reference to ‘learning styles’  is made: Since evidence suggests that learning styles are a scientific myth, I would try to avoid this term, and only describe this as a student preference (as the authors already do on line 1197). It is also somethin worth mentioning in the discussion section that the myth of learning styles is persistent. 

Discussion

The findings are discussed in a rich and cross-connective manner, making it a interesting read.

13. However, the final parts of the discussion (potential implications & conclusion) seem a bit repetitive here and there and could be written a bit more concise.

14. The discussion starts with a description of an hierarchical order of the tree types of support, and it is mentioned that this is reflected in the findings from the interviews. However, it is not clarified how this is reflected in the findings. This needs to be elucidated further. Also, I would have expected a discussion of the more specific findings first, based on what was identified as the primary research question. Perhaps this hierarchical order fits better with a more overarching discussion of findings later in the discussion section.

15. On page 32, line 1392-1983, the authors refer to Figure 3, but I think this should be a reference to Table 3?

16. The paragraph on Page 33 is quite long, which makes it harder to read. This could be divided into two paragraphs, for example where the discussion of indirect support strategies begins (line 1428).

17. On page 33, line 1441, it is mentioned that certain strategies do not seem to address WM challenges, but it is not immediately clear if this conclusion is based on the finding that teachers do not mention these strategies. It is for example stated that connections to the real world do not seem to alleviate WM demands. However, some strategies that offer connections to the real world, such as using real materials, or providing context, or using embodied approaches might also provide students with support or build schemata that help to reduce cognitive load (see for example: Pouw, van Gog, & Paas, 2014).

18. On page 34, line 1456, the findings regarding varying modalities and materials is discussed as part of classroom organization, but could also be linked to instructional support. The multimedia theory or dual coding theory – which is related to cognitive load theory – is relevant to mention in light of this finding as well.

 References:

Kahl, T., Segerer, R., Grob, A., & Möhring, W. (2022). Bidirectional associations among executive functions, visual-spatial skills, and mathematical achievement in primary school students: Insights from a longitudinal study. Cognitive Development62, 101149.

Peng, P., & Kievit, R. A. (2020). The development of academic achievement and cognitive abilities: A bidirectional perspective. Child Development Perspectives14(1), 15-20.

Pouw, W. T., Van Gog, T., & Paas, F. (2014). An embedded and embodied cognition review of instructional manipulatives. Educational Psychology Review26, 51-72.

Author Response

Please see the attachment for the point-by-point response to the reviewer's comments.

Reviewer 2 Report

Comments and Suggestions for Authors

Comments to the authors

This manuscript focuses on the perceptions of teachers in schools in Belgium and the Netherlands around their pupils’ working memory and the activities that affect that.

Thank you for the opportunity to review this manuscript. I very much enjoyed reading this manuscript and I think it has potential to contribute to the literature in several ways, especially around working memory and the role of educational practitioners.

The abstract is excellent and very clearly outlined the aims of this study, as well as its key elements in terms of method and key findings.

Introduction

The relevant literature review is of significant depth, and it clearly provides an overview of the key areas that this study is based upon. Each section is detailed and clear with links to all the relevant authors and studies that I would expect around WM, which is a key strength of this manuscript in terms of relevance and value for other researchers and practitioners. I appreciate the inclusion of the visualisation for the Teaching Through Interactions framework. It really helped highlight what had already been outlined in terms of relevant literature for this study.

The aims and research questions for this current study were presented in a clear way at the end of this section.

Materials and Methods

This section provided even further strengths to this manuscript. The participant recruitment process, as well as the details of the participants themselves were well-defined, which I very much appreciated.

The procedure and tools in terms of data collection were also presented in a clear and detailed manner. It was very clear to me what the participants were asked to do during their interviews.

The steps followed in terms of the data analysis were incredibly detailed with details for each step of the thematic analysis followed. The authors have done an excellent job with this section!

Results

This section outlined the key themes from the data, which were presented in a well-structured and clear way. I particularly appreciated the inclusion of several direct quotes from participants which helped further emphasise the significance of each theme and sub-theme.

Discussion

This section continues the excellent series of strengths of this manuscript. Continuing from the Results section, the findings have been linked very closely with previous relevant studies around WM, teaching strategies and the overall role of teachers. The aims of the current study were successfully met, and the authors have highlighted that in this section. Table 3 was an excellent addition in terms of visualisation of the findings, as it very transparently showed the links between the finding and the TTI framework.

The limitations section was transparent without undermining the strengths of this study. I also particularly appreciated the details recommendations for practitioners in the potential implications section.

Referencing:

The numbering of citations in the whole manuscript requires significant editing as the citations throughout should be numbered as [1] (instead of the authors’ names). The referencing guidance on the MDPI can be very helpful with that.

Author Response

Dear Reviewer 2,

We would like to express our gratitude for taking the time to thoroughly review our manuscript.

We are pleased to hear that you found our study interesting and recognise the potential contribution to the literature, particularly in the realm of working memory and the role of teachers in supporting students in the classroom.

Your recognition of the depth and clarity of the literature review presented is encouraging. We strive for transparency and clarity in our research methodology, and we are glad that our efforts in this regard have met your expectations. We are pleased that you found the presentation of key findings to be well-structured, clear, and linked with outlined theories. Finally, we are glad that you found our aims to be successfully met and recommendations for practitioners in the implications section clearly stated.

We took your feedback regarding the citation numbering into account and have revised the manuscript accordingly. We have updated all citations in the manuscript to adhere to the numerical format, as per the guidelines, and have sorted the reference list accordingly.

We thank you for your time, effort, and expertise.